# Circadian regulation of endoplasmic reticulum calcium response in cultured mouse astrocytes

**Ji Eun Ryu[1,2†], Kyu-Won Shim[3†], Hyun Woong Roh[1,4†], Minsung Park[1,2], Jae-Hyung Lee[5]\*, Eun Young Kim[1,2]\***

[1]Neuroscience Graduate Program, Department of Biomedical Sciences, Ajou University Graduate School of Medicine, Suwon, Republic of Korea; [2]Department of Brain Science, Ajou University School of Medicine, Suwon, Republic of Korea; [3]Interdisciplinary Program in Bioinformatics, Seoul National University, Seoul, Republic of Korea; [4]Department of Psychiatry, Ajou University School of Medicine, Suwon, Republic of Korea; [5]Department of Oral Microbiology, College of Dentistry, Kyung Hee University, Seoul, Republic of Korea

**\*For correspondence:**
jaehlee@khu.ac.kr (JHL);
ekim@ajou.ac.kr (EYK)

[†]These authors contributed equally to this work

**Competing interest:** The authors declare that no competing interests exist.

## eLife Assessment

This work describes a circadian regulation in the expression of HERP, a regulator of endoplasmic reticulum calcium, in primary astrocytic cultures. This work is **important** because it highlights the potential importance of circadian rhythms in astrocytes, even though making a direct comparison between these rhythms in vitro and in vivo remains challenging. The technical approaches used in this work (RNA-seq, siRNA, Ca2+ imaging) are a **solid** support for data interpretation.

**Abstract** The circadian clock, an internal time-keeping system orchestrates 24 hr rhythms in physiology and behavior by regulating rhythmic transcription in cells. Astrocytes, the most abundant glial cells, play crucial roles in CNS functions, but the impact of the circadian clock on astrocyte functions remains largely unexplored. In this study, we identified 412 circadian rhythmic transcripts in cultured mouse cortical astrocytes through RNA sequencing. Gene Ontology analysis indicated that genes involved in $Ca^{2+}$ homeostasis are under circadian control. Notably, *Herpud1* (*Herp*) exhibited robust circadian rhythmicity at both mRNA and protein levels, a rhythm disrupted in astrocytes lacking the circadian transcription factor, BMAL1. HERP regulated endoplasmic reticulum (ER) $Ca^{2+}$ release by modulating the degradation of inositol 1,4,5-trisphosphate receptors (ITPRs). ATP-stimulated ER $Ca^{2+}$ release varied with the circadian phase, being more pronounced at subjective night phase, likely due to the rhythmic expression of ITPR2. Correspondingly, ATP-stimulated cytosolic $Ca^{2+}$ increases were heightened at the subjective night phase. This rhythmic ER $Ca^{2+}$ response led to circadian phase-dependent variations in the phosphorylation of Connexin 43 (Ser368) and gap junctional communication. Given the role of gap junction channel (GJC) in propagating $Ca^{2+}$ signals, we suggest that this circadian regulation of ER $Ca^{2+}$ responses could affect astrocytic modulation of synaptic activity according to the time of day. Overall, our study enhances the understanding of how the circadian clock influences astrocyte function in the CNS, shedding light on their potential role in daily variations of brain activity and health.

## Introduction

The circadian clock orchestrates a range of behaviors, physiological processes, and cellular and biochemical activities in organisms with approximately 24 hr rhythms, facilitating adaptation to environmental changes (*Bell-Pedersen et al., 2005*). In mammals, the master clock localized in the suprachiasmatic nucleus (SCN) of the anterior hypothalamus synchronizes peripheral tissue clocks throughout the body to the light/dark cycle (*Mohawk et al., 2012*). The circadian clock's molecular mechanism is driven by a transcriptional translational feedback loop (TTFL). CLOCK and BMAL1 form a heterodimer that triggers expression of the repressor genes, *Period* (*Per*) and *Cryptochrome* (*Cry*), along with other clock-controlled output genes. PER and CRY proteins then translocate to the nucleus, where they interact with CLOCK/BMAL1, halting further transcriptional activation. A secondary feedback loop involving the interplay between transcriptional activation by ROR and repression by REV-ERB finely regulates the rhythmic transcription of *Bmal1* (*Takahashi, 2017*). Recent studies have revealed that the circadian clock controls the rhythmic transcription of numerous genes, displaying notable tissue specificity. This characteristic underscores the time-of-day-specific functional specialization of diverse tissues (*Mure et al., 2018*; *Zhang et al., 2014*). Consequently, circadian clock-controlled genes significantly influence various physiological processes including development, metabolism, aging, and neurodegeneration (*Lananna and Musiek, 2020*; *Reinke and Asher, 2019*; *Schultz and Kay, 2003*).

Astrocytes, the most abundant glial cells in the CNS, play vital roles in maintaining homeostasis of extracellular fluids, ions, and neurotransmitters; proving energy substrates to neurons; regulating local blood flow; modulating neuronal activity through gliotransmission; and assisting in interstitial fluid drainage (*Sofroniew and Vinters, 2010*). Although not electrically excitable, astrocytes exhibit complex intracellular $Ca^{2+}$ signaling, both spatially and temporally, within individual astrocytes and across astrocytic networks. Astrocytes express various types of Gq-coupled receptors activated by neurotransmitters such as ATP, epinephrine, glutamate, and dopamine *etc.* (*Verkhratsky and Nedergaard, 2018*). Activation of these Gq-coupled receptors initiates the inositol 1,4,5-trisphosphate receptor ($IP_3R$) signaling cascade, causing a release of $Ca^{2+}$ from the ER that substantially increases intracellular $Ca^{2+}$ levels (*Guerra-Gomes et al., 2017*; *Verkhratsky and Nedergaard, 2018*). Increases in intracellular $Ca^{2+}$ in astrocytes are involved in regulating numerous afore-mentioned processes (*Giaume et al., 2021*; *Verkhratsky and Nedergaard, 2018*). Recent studies have shown that astrocyte $Ca^{2+}$ signaling can regulate circadian rhythms and sleep patterns (*Bojarskaite et al., 2020*; *Brancaccio et al., 2019*).

Accumulating evidence underscores the importance of the circadian clock in astrocytes function. Cultured murine cortical astrocytes expressing a PER-luciferase reporter (PER::LUC) exhibit circadian oscillations and can be entrained by temperature cycle (*Prolo et al., 2005*) and vasoactive intestinal polypeptide (VIP), which synchronizes SCN (*Marpegan et al., 2009*). These findings provide strong evidence that a molecular clock operates in astrocytes. In addition, disruption of clock genes specific to astrocytes has been linked to a range of deleterious effects, including excessive astrocyte activation, disruptions in daily activity patterns, impaired learning, metabolic imbalances, and a shortened lifespan (*Barca-Mayo et al., 2020*; *Barca-Mayo et al., 2017*; *Griffin et al., 2019*; *Lananna et al., 2018*). Remarkably, these phenotypic changes closely resemble those observed in astrocytes during the progression of neurodegenerative diseases (*Brandebura et al., 2023*; *Phatnani and Maniatis, 2015*). Moreover, the rhythmic metabolic activity of astrocytes, including intracellular $Ca^{2+}$ fluctuations and neurotransmitter release, is critical for circadian timing in the SCN (*Brancaccio et al., 2017*). Despite the emerging significance of circadian rhythms in astrocyte function in health and disease, our understanding of the link between the circadian clock and specific astrocytic functions is still limited.

In this study, we conducted a transcriptome analysis to investigate how the circadian clock regulates astrocyte function. Our research uncovered that *Herp* (homocysteine-inducible ER protein with ubiquitin like domain 1; *Herp*) exhibited a BMAL1-dependent rhythmic expression, peaking at subjective midday phase and reaching its nadir at subjective midnight phase. HERP was crucial for ER $Ca^{2+}$ regulation by modulating the degradation of ITPRs. Importantly, the ATP-stimulated ER $Ca^{2+}$ response varied according to time post synchronization, a dependence that was absent in cultured astrocytes from *Bmal1*$^{-/-}$ mice. Furthermore, the pronounced release of $Ca^{2+}$ during the subjective night phase led to robust phosphorylation of the Ser368 residue of CX43, diminishing cell-to-cell communication between astrocytes. Overall, our study demonstrates that the circadian clock orchestrates a variety

of astrocytic processes including $Ca^{2+}$ homeostasis by controlling the rhythmic transcription of genes, highlighting its pivotal role in CNS function.

## Results

### Defining circadian rhythmic transcripts in cultured mouse cortical astrocytes

To understand how the circadian clock regulates astrocyte physiology, we conducted a circadian transcriptome analysis of cultured mouse cortical astrocytes. Cultured astrocytes were synchronized using serum shock (SS) and harvested every 4 hr for 2 days starting at 12 hr post-SS. The time series astrocyte transcriptome was determined by RNA-seq (*Figure 1A*). Using a Gaussian mixture model, we defined expressed transcripts, choosing 0.577 transcripts per million (TPM) as the cut-off, corresponding to the 1% threshold in the distribution curve of highly expressed genes (*Figure 1—figure supplement 1A*). A total of 17,671 transcripts with the highest TPM values of the 12 time points exceeding 0.577, were considered expressed. To validate astrocyte enrichment, we compared the expression of marker genes for nervous cells. Our transcriptome data were replete with astrocyte markers but lacked markers for microglia, oligodendrocytes, neurons, and endothelial cells (*Figure 1—figure supplement 1B*).

Various methods have been used to identify periodicity in time-series data, such as Lomb-Scargle (*Glynn et al., 2006*), JTK_CYCLE (*Hughes et al., 2010*), and ARSER (*Yang and Su, 2010*), each with distinct advantages and limitations. MetaCycle, integrates these three methods, facilitating the evaluation of periodicity in time-series data without requiring the selection of an optimal algorithm (*Wu et al., 2016*). Additionally, BioCycle has been developed using a deep neural network trained with extensive synthetic and biological time series datasets (*Agostinelli et al., 2016*). Because MetaCycle and Biocycle identify periodic signal based on different algorithms, we applied both packages to identify periodicity in our time-series transcriptome data. BioCycle and MetaCycle analyses detected 321 and 311 periodic transcripts, respectively (FDR corrected, q-value <0.05; *Figure 1B*). Among these, 220 (53.4%) were detected by both methods, but many transcripts did not overlap. MetaCycle is known for its inability to detect asymmetric waveforms (*Mei et al., 2021*). In our analysis, genes with increasing waveforms like *Adora1* and *Mybph* were identified as rhythmic only by BioCycle, while *Plat* and *Il34* were identified as rhythmic only by MetaCycle (*Figure 1—figure supplement 1C*). Despite these discrepancies, the clear circadian rhythmic expression profiles of these genes led us to conclude that using the union of the two lists compensates for the limitations of each algorithm. Consequently, we identified a total of 412 circadian rhythmic transcripts (2.3% of all transcripts; *Supplementary file 1*). The circadian oscillations of these transcripts are illustrated in the heatmap in *Figure 1—figure supplement 1D*.

Next, we compared rhythmic transcripts of astrocytes with those of 12 mouse tissues from CircaDB (*Zhang et al., 2014*), a database of circadian rhythmic genes (*Figure 1C*). A majority of the rhythmic transcripts in astrocytes (265 of 412, 65.3%) were not rhythmic or were rhythmic in one tissue. 14 transcripts were rhythmic in more than 10 tissues; most of these were core clock genes except *Tspan4*, *Tsc22d3*, and *Wee1*. This result is consistent with previous reports of tissue-specific circadian rhythmic expression (*Mure et al., 2018*; *Panda et al., 2002*). A phase comparison of these 14 common oscillating transcripts between CircaDB and our analysis showed a consistent time difference, with 8 hr after SS corresponding to ZT0 (*Figure 1D*). An additional phase comparison of 82 transcripts that overlapped in only one tissue indicated a very robust phase correlation between astrocytes and the corresponding tissue (r=0.69, p<0.001; *Figure 1E*). Collectively, our analysis indicates that the identified oscillating transcripts were not false positive but were clearly associated with the circadian clock. Hierarchical clustering analysis of oscillating transcripts revealed two clusters: cluster 1 with 185 transcripts and cluster 2 with 227 transcripts (*Figure 1—figure supplement 2*). A circular map of peak phases across all rhythmic genes revealed two major peaks, one in the early night (ZT12 to ZT16) and one in the late night or dawn (ZT21 to ZT1; *Figure 1F*). Here for more intuitive interpretation, we set 8 hr post synchronization (sync) to ZT0 based on a previous phase comparison. Circadian transcriptome studies conducted in multiple animal tissues have indicated that the phases of each tissue are not randomly distributed but are predominantly clustered in one or two narrow temporal windows (*Panda et al., 2002*). Consistent with these previous reports, mouse cultured astrocytes exhibited

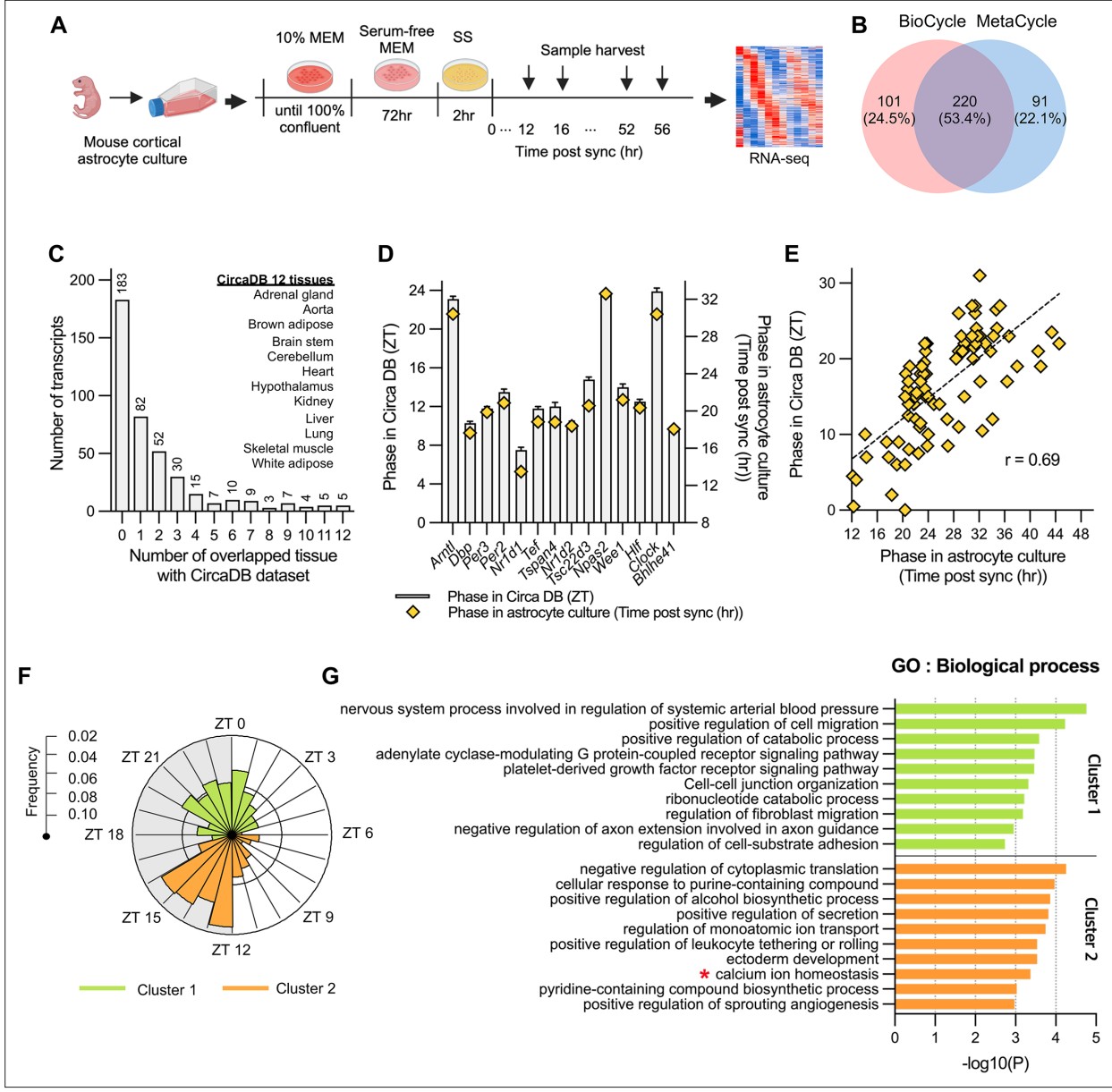

**Figure 1.** Circadian rhythmic transcripts in mouse cultured cortical astrocytes. (**A**) Experimental scheme for synchronizing circadian rhythms in mouse cultured cortical astrocytes followed by RNA-sequencing over 2 days. (**B**) Venn diagram displaying the number of circadian rhythmic transcripts identified by two algorithms (q<0.05 in MetaCycle or BioCycle). (**C**) Number of transcripts that overlapping with 12 tissues in mouse circadian transcriptome ss from CircaDB (http://circadb.hogeneschlab.org). (**D**) Comparison of mean phase (ZT) from CircaDB and peak phase (Time post sync) from cultured astrocytes for 14 transcripts that are rhythmic in 10 or more tissues. (**E**) Scatter plot showing phase in CircaDB and peak phase of cultured astrocytes for transcripts that are rhythmic in one tissue. (**F**) Radial histogram of the distribution of phases of rhythmic genes in the astrocyte transcriptome. (**G**) Top 10 enriched GO Biological Process (BP) terms for significant circadian rhythmic genes in astrocytes (p<0.01) identified by Metascape (https://metascape.org). Panel A was created with BioRender.com.

The online version of this article includes the following figure supplement(s) for figure 1:

**Figure supplement 1.** Identification of circadian rhythmic transcripts in mouse cultured astrocytes.

**Figure supplement 2.** Cluster analysis of circadian rhythmic transcripts.

a high degree of temporal organization. Intriguingly, there was a relatively quiescent zone during subjective daytime, a finding different from that observed for diurnal animals, which showed a quiescent zone during subjective nighttime (*Mure et al., 2018*).

To understand the biological processes and pathways controlled by the circadian clock, we performed a Gene Ontology (GO) analysis on each cluster using Metascape (https://metascape.org; *Zhou et al., 2019*; *Figure 1G*). Given that the goal of our research was to investigate processes specifically controlled by the circadian clock, we excluded 17 core clock transcripts (*Arntl, Clock, Nfil3, Npas2, Bhlhe40, Bhlhe41, Cry1, Cry2, Dbp, Elf, Nr1d1, Nr1d2, Per1, Per2, Per3, Rorc*, and *Tef*) from our GO enrichment analysis. Among the rhythmic processes, we focused on 'calcium ion homeostasis (GO:0055074)' due to the importance of intracellular $Ca^{2+}$ signaling in numerous astrocyte functions (*Agulhon et al., 2008*; *Bojarskaite et al., 2020*; *Brancaccio et al., 2019*; *Giaume et al., 2021*; *Guerra-Gomes et al., 2017*; *Verkhratsky and Nedergaard, 2018*). In this category, $Ca^{2+}$ homeostasis-related genes included *Herp, Slc4a11, Sord* and *Kcnh1 etc.* with *Herp* showing the most robust oscillation with large amplitude (*Supplementary file 2A*).

## Circadian rhythmic expression of *Herp* in cultured mouse cortical astrocytes

We first assessed the expression patterns of core clock genes and *Herp* in our cultured astrocytes. *Bmal1*, was rhythmically expressed with a peak at 30 hr post sync while *Per2* and *Rev-Erbα* showed rhythmic expression with peaks at 20 hr and 16 hr, respectively (*Figure 2A*). *Herp* displayed a robust rhythmic expression pattern, peaking at 20 hr, similar to *Per2* (*Figure 2A*). To verify the circadian expression of *Herp*, we performed quantitative real-time RT-PCR, which confirmed that *Bmal1*, *Rev-Erbα* and *Herp* exhibited circadian rhythms consistent with the RNAseq results (*Figure 2B*).

Next, we examined whether HERP protein levels also showed rhythmic changes, as many genes exhibit rhythmic transcription without corresponding protein level changes (*Reddy et al., 2006*). We validated the specificity of HERP antiserum using Western blot analysis of astrocytes treated with *Herp* siRNA (*Herp*-KD), which significantly reduced *Herp* mRNA levels (*Figure 2C*) and the intensities of specific bands in western blots (*Figure 2D and E*). SS-synchronized cultured astrocytes were sampled every 6 hr for 2 days and processed for western blot analysis of BMAL1 and HERP. BMAL1 phosphorylation showed circadian variation, with hyper-phosphorylation peaking at midday and hypo-phosphorylation peaking early in the day or late at night (*Yoshitane et al., 2009*). In our SS-synchronized cultured astrocytes, BMAL1 was hyperphosphorylated at 18 hr, 24 hr, 42 hr, and 48 hr (corresponding to midday) and hypo-phosphorylated at 12 hr, 36 hr (corresponding to early in the day), and 30 hr, 54 hr (corresponding to late at night) throughout two daily cycles (*Figure 2F*) indicating strong circadian rhythms in cultured astrocytes. Importantly, HERP protein levels oscillated peaking at 18 hr and 42 hr and reaching a minimum at 12 hr, 30 hr, and 54 hr, mirroring its mRNA pattern (*Figure 2F and G*).

*Herp* also exhibited rhythmic expression in astrocyte cultures synchronized by either dexamethasone (*Balsalobre et al., 2000*) or forskolin (*Yagita and Okamura, 2000*) treatments which are other commonly used for circadian clock resetting (*Figure 2—figure supplement 1*). Finally, to investigate whether the rhythmic expression of *Herp* is regulated by a circadian clock, we examined *Herp* mRNA and protein expression patterns in primary astrocyte cultured from *Bmal1$^{-/-}$* mice. The rhythmic expression patterns of *Per2*, *Rev-Erbα*, and *Herp* were abolished in *Bmal1$^{-/-}$* astrocyte cultures with their expression maintained at trough levels (*Figure 2A*). HERP protein levels remained constant throughout the daily cycles (*Figure 2H and I*). Collectively, these observations indicate that the expression of *Herp* exhibits a robust circadian rhythm that is controlled by BMAL1 in cultured mouse astrocytes. These results suggest that HERP-regulated cellular processes vary in a time-of-day specific manner.

## *Herp* knockdown alters ATP-induced ER Ca$^{2+}$ release

*Herp* was first identified as a gene with altered expression in response to homocysteine treatment in human umbilical vein endothelial cells (HUVECs; *Kokame et al., 1996*). It is strongly induced not only by homocysteine but also by ER stress-causing agents such as tunicamycin or thapsigargin. HERP, characterized by an N-terminal ubiquitin-like domain and present in the ER membrane facing cytoplasmic side, is involved in the unfolded protein response (UPR; *Kokame et al., 2000*) and is part of the ER-associated degradation (ERAD) complex, participating in the ubiquitination and relocation of

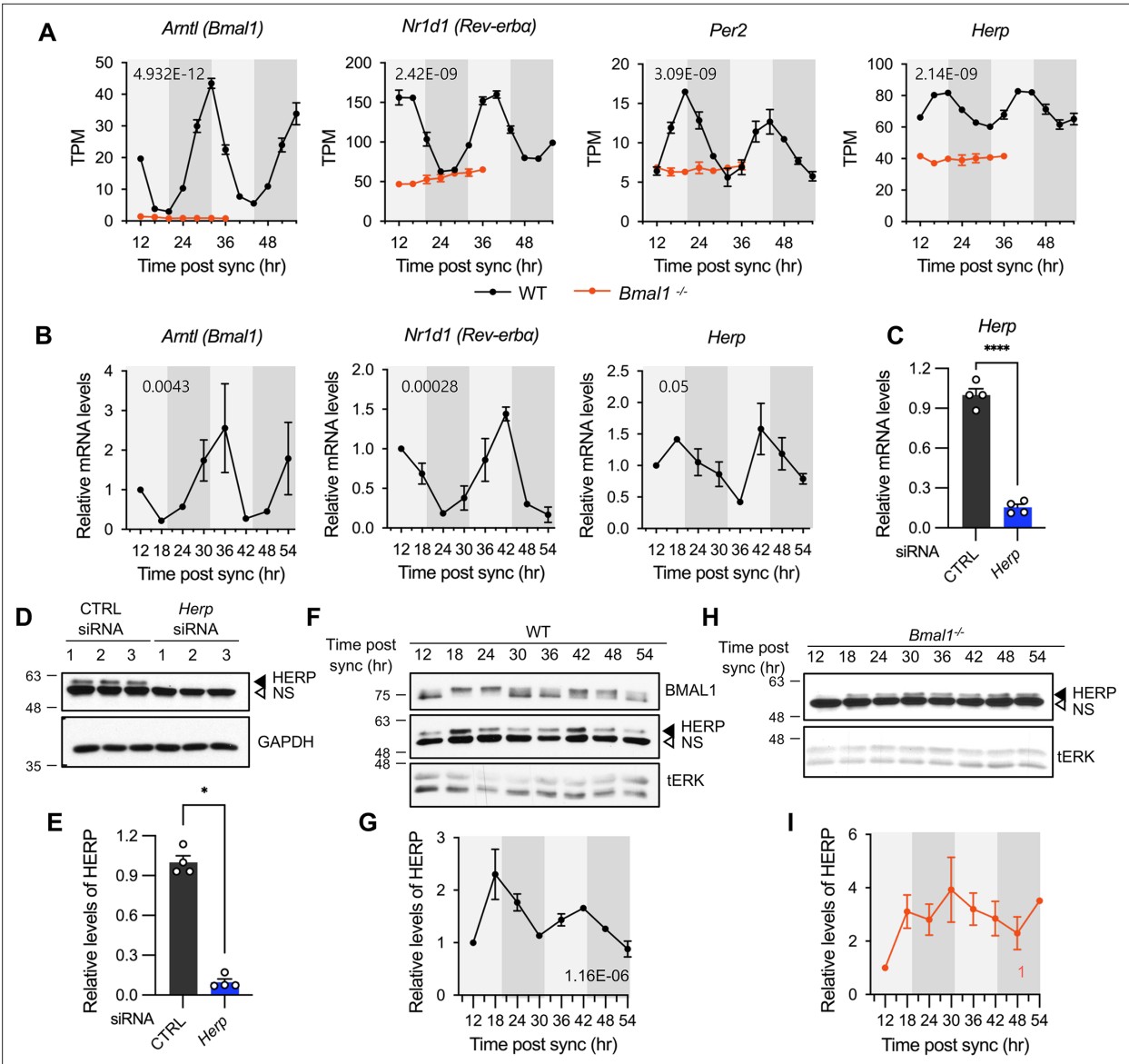

**Figure 2.** *Herp* is rhythmically expressed in mouse cultured astrocytes and its expression is controlled by BMAL1. (**A, B**) Astrocyte cultures from rom WT (black) and *Bmal1⁻/⁻* (orange) mice were synchronized and harvested at the indicated time. Expression was analyzed for rhythmicity using MetaCycle, and expression levels of given genes were quantified using RNA-seq (**A**) and real-time qRT-PCR (**B**) data. Values are mean ± SEM (n=2; p-values are indicated by insets in graphs). Light gray and dark gray backgrounds represent subjective day and subjective night, respectively, based on the phase analysis from **Figure 1D**. (**C–E**) Cultured astrocytes were transfected with the indicated siRNA (20 nM) and processed for real-time qRT-PCR (**C**) and western blot (**D, E**) analyses at 48 hrs post transfection. (**C**) Values are mean ± SEM (n=4; *p<0.05, **p<0.005, ***p<0.0005, and ****p<0.00005; *t*-test). (**D**) Representative western blot images of four independent experiments are shown. GAPDH served as loading control. (**E**) Densitometric quantification of HERP levels, normalized to GAPDH levels. Values are mean ± SEM (n=4; ***p<0.0005; Mann-Whiteny *U* test). (**F–I**) Astrocyte cultures from rom WT (**F, G**) and *Bmal1⁻/⁻* (**H, I**) mice were synchronized and harvested at the indicated time for western blot analysis. Total ERK (tERK) served as a loading control. HERP/tERK values at different times were normalized to those at 12 hr post sync (set to 1). Meta2d p values are indicated by insets in graphs. Light gray and dark gray backgrounds represent subjective day and subjective night, respectively. (**F**) Representative western blot images from five independent experiments are shown. (**H**) Representative western blot images from two independent experiments are shown.

The online version of this article includes the following source data and figure supplement(s) for figure 2:

**Source data 1.** PDF file containing original western blot for **Figure 2D**, indicating the relevant bands and treatments.

**Source data 2.** Original files for western blot analysis displayed in **Figure 2D**.

**Source data 3.** PDF file containing original western blot for **Figure 2F**, indicating the relevant bands and treatments.

**Source data 4.** Original files for western blot analysis displayed in **Figure 2F**.

*Figure 2 continued on next page*

*Figure 2 continued*

**Source data 5.** PDF file containing original western blot for *Figure 2H*, indicating the relevant bands and treatments.

**Source data 6.** Original files for western blot analysis displayed in *Figure 2H*.

**Figure supplement 1.** Herp is rhythmically expressed in mouse cultured astrocytes synchronized by forskolin or dexamethasone.

**Figure supplement 1—source data 1.** PDF file containing original western blot for *Figure 2—figure supplement 1F*, indicating the relevant bands and treatments.

**Figure supplement 1—source data 2.** Original files for western blot analysis displayed in *Figure 2—figure supplement 1F*.

**Figure supplement 1—source data 3.** PDF file containing original western blot for *Figure 2—figure supplement 1D*, indicating the relevant bands and treatments.

**Figure supplement 1—source data 4.** Original files for western blot analysis displayed in *Figure 2—figure supplement 1D*.

ERAD substrates (*Leitman et al., 2014*). Notably, HERP modulates the ER $Ca^{2+}$ response through $IP_3R$ degradation (*Paredes et al., 2016*; *Torrealba et al., 2017*).

Thus, we sought to investigate whether HERP-controlled processes are subject to circadian regulation by examining the HERP-regulated ER $Ca^{2+}$ response in the astrocytes. We employed the organelle-specific fluorescent $Ca^{2+}$ indicators: G-CEPIA1er for the ER (*Suzuki et al., 2014*), R-GECO1 for the cytosol (*Wu et al., 2013*), and mito-R-GECO1 for mitochondria (*Zhao et al., 2011*). We first confirmed that these $Ca^{2+}$ indicators colocalized with organelle-specific markers (*Figure 3—figure supplement 1*). ATP serves not only as a fundamental energy source but also as an active intercellular messenger. In astrocytes, ATP application induces an increase in intracellular $Ca^{2+}$ (*Neary et al., 1988*) by binding to the $G_q/G_{11}$ coupled P2Y receptor, activating PLC-β and inducing the hydrolysis of $PIP_2$ to diacylglycerol (DAG) and $IP_3$, which mobilizes $Ca^{2+}$ through $IP_3Rs$ on the ER membrane (*Baryshnikov et al., 2003*; *James and Butt, 2001*).

We treated control and *Herp*-KD astrocytes with 100 μM ATP and monitored subcellular $Ca^{2+}$ changes. ATP treatment rapidly decreased ER $Ca^{2+}$ in control astrocytes and this response was more pronounced in *Herp*-KD astrocytes (*Figure 3A–C*). As the main $Ca^{2+}$ store in the cell, ER releases $Ca^{2+}$ which is then rapidly transmitted to other organelles such as mitochondria and cytosol (*Carreras-Sureda et al., 2018*; *Giorgi et al., 2018*). Although ATP treatment did not significantly alter cyto-solic $Ca^{2+}$ signal in control astrocytes, it greatly increased cytosolic $Ca^{2+}$ signals in *Herp*-KD astrocytes (*Figure 3D–F*). Consistently, mitochondrial $Ca^{2+}$ significantly increased in *Herp*-KD astrocytes compared with control (*Figure 3G–I*).

Next, we examined the mechanism by which HERP controls ER $Ca^{2+}$. There are three subtypes of $IP_3Rs$ encoded by the genes *Itpr1*, *Itpr2*, and *Itpr3*. Transcriptome analysis of our cultured astro-cytes indicated *Itpr1* as the most prevalent, followed by *Itpr2*, while *Itpr3* expression was minimal (*Figure 3—figure supplement 2*). Notably, expression of these *Itpr* subtypes was not rhythmic. We then assessed the ITPR1 and ITPR2 protein levels in control and *Herp*-KD astrocytes and found that the levels of both ITPR1 and ITPR2 were slightly but statistically significantly increased in *Herp*-KD astro-cytes compared to controls (*Figure 3J–M*). These results, consistent with previous reports (*Paredes et al., 2016*; *Torrealba et al., 2017*) indicated that HERP negatively regulates ITPRs in astrocyte.

To confirm that the HERP-ITPR axis underlies ER $Ca^{2+}$ response, we analyzed the ER $Ca^{2+}$ response following treatment with Xestospongin C (XesC), an IP3R inhibitor. XesC treatment reduced the ATP-induced ER $Ca^{2+}$ release and abolished the enhanced effect observed in *Herp*-KD compared to control (*Figure 3N–P*). Collectively, these results clearly indicate that HERP controls ER $Ca^{2+}$ release by nega-tively regulating IP3Rs.

## ATP-induced ER $Ca^{2+}$ release varies according to time post synchronization

Given that HERP levels exhibited circadian variation and HERP regulated ER $Ca^{2+}$ release, we inves-tigated whether ATP-induced ER $Ca^{2+}$ responses differ depending on circadian phase. After SS, we measured ER $Ca^{2+}$ following ATP treatment at peak (42 hr post sync) and trough (30 hr post sync) phases of HERP (*Figure 4A*). With lower HERP levels potentially resulting in higher $IP_3R$ levels at 30 hr, we anticipated greater ER $Ca^{2+}$ release at 30 hr than at 42 hr. Consistent with this, ER $Ca^{2+}$ decreased more at 30 hr than at 42 hr upon ATP treatment (*Figure 4B–D*). Also, in keeping with previous result (*Figure 3A–I*), cytosolic $Ca^{2+}$ increased more at 30 hr than at 42 hr following ATP

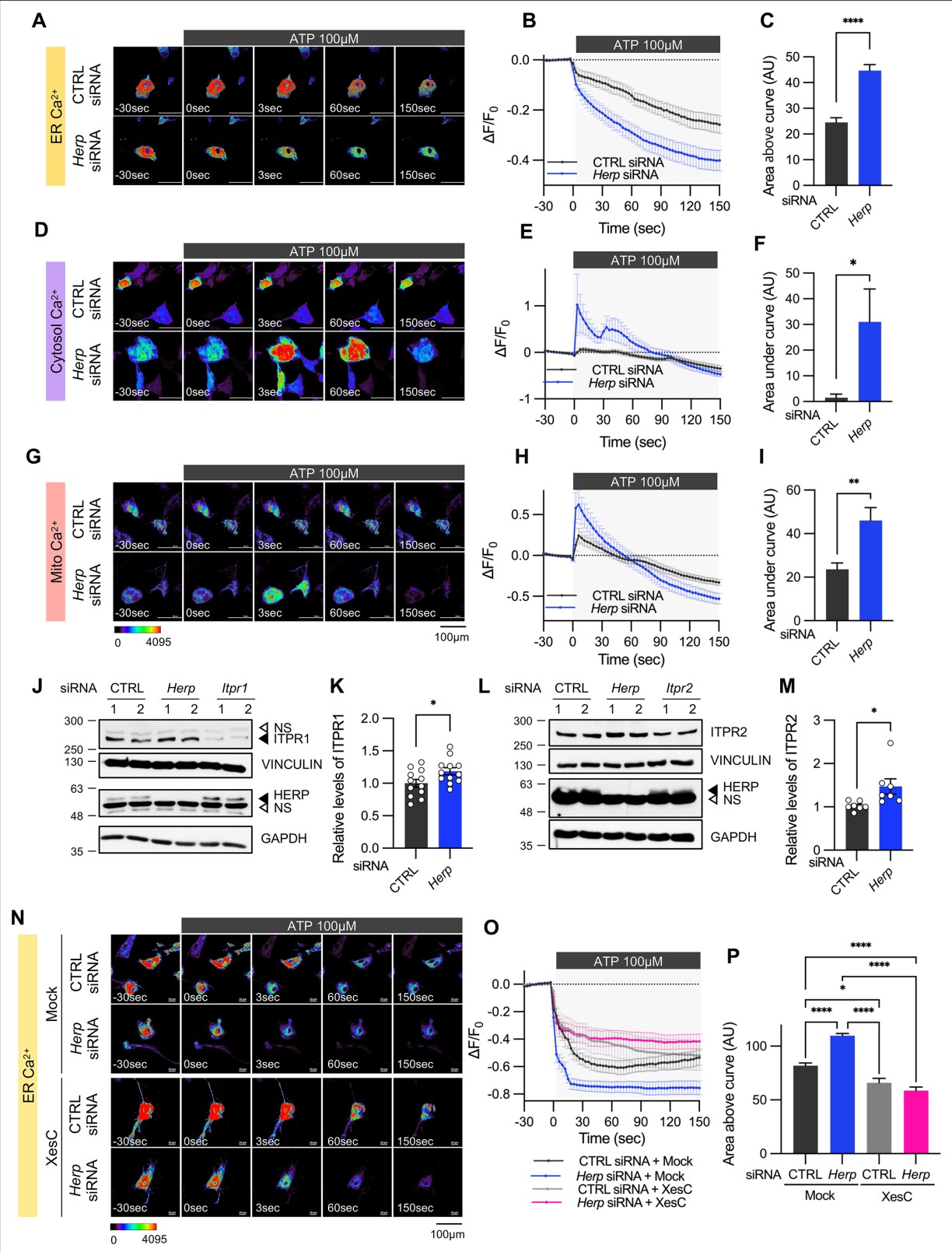

**Figure 3.** *Herp* knockdown altered ATP-induced ER Ca²⁺ response. (**A–I**) Cultured astrocytes were co-transfected with 20 μM non-targeting (CTRL) siRNA or *Herp* siRNA together with G-CEPIA1er (**A–C**), R-GECO1 (**D–F**) or mito-R-GECO1 (**G–I**). At 48 hr post transfection, cultured astrocytes were treated with 100 μM ATP and Ca²⁺ imaging analysis was performed. Images were acquired every 3 seconds. (**A, D, G**) Representative time-lapse images of each Ca²⁺ indicator. (**B, E, H**) ΔF/F₀ values over time following ATP application. (**C, F, I**) Area above or area under the curve values, calculated from

*Figure 3 continued on next page*

*Figure 3 continued*

panels B, E, and H. (**A–C**) CTRL siRNA, n=19; *Herp* siRNA, n=22. (**D–F**) CTRL siRNA, n=20; *Herp* siRNA, n=25. (**G–I**) CTRL siRNA, n=16; *Herp* siRNA, n=16. (**J–M**) Cultured astrocytes were transfected with the indicated siRNA (20 nM) and processed for Western blot analysis 48 hr post transfection. Vinculin and GAPDH served as loading control for ITPRs and HERP, respectively. (**J**) Representative western blot images from twelve independent experiments are shown. NS, non-specific band (**K**) Densitometric quantification of western blot data showing relative levels of ITPR1 in *Herp* siRNA-transfected astrocytes compared to CTRL siRNA transfected astrocytes. (**L**) Representative Western blot images from five independent experiments. (**M**) Densitometric quantification of western blot data showing relative levels of ITPR2 in *Herp* siRNA-transfected astrocytes compared to Control astrocytes. Values are mean ± SEM (*p<0.05, **p<0.005, ***p<0.0005, ****p<0.00005; t-test). (**N–P**) Cultured astrocytes were treated with 10 µM Xestospongin C (XesC), an IP3R inhibitor, for 30 min before live imaging. Cells were then treated with 100 µM ATP, and images were captured every 3 s. (**N**) Representative time-lapse images of ER $Ca^{2+}$ indicator. (**O**) $\Delta F/F_0$ values over time following ATP application. (**P**) Area above the curve values were calculated from panel O. CTRL siRNA + Mock, n=9; *Herp* siRNA + Mock, n=9; CTRL siRNA + XesC, n=8; *Herp* siRNA + XesC, n=14. Values are means ± SEM (*p<0.05, **p<0.005, ***p<0.0005, ****p<0.00005; one-way ANOVA).

The online version of this article includes the following source data and figure supplement(s) for figure 3:

**Source data 1.** PDF file containing original western blot for *Figure 3J*, indicating the relevant bands and treatments.

**Source data 2.** Original files for western blot analysis displayed in *Figure 3J*.

**Source data 3.** PDF file containing original western blot for *Figure 3L*, indicating the relevant bands and treatments.

**Source data 4.** Original files for western blot analysis displayed in *Figure 3L*.

**Figure supplement 1.** Co-localization of organelle-specific $Ca^{2+}$ sensors with organelle markers.

**Figure supplement 2.** Expression profiles of Itpr1, Itpr2 and Itpr3 in cultured astrocytes from RNA-seq data.

---

treatment (*Figure 4E–G*). Although mitochondrial $Ca^{2+}$ responses exhibited a similar trend, the differences were not statistically significant (*Figure 4H–J*).

To confirm that HERP underlies this circadian phase-dependent ER $Ca^{2+}$ release, we compared ER $Ca^{2+}$ release in control and *Herp*-KD astrocytes at 30 hr and 42 hr post sync. Control astrocytes exhibited time-dependent variation in ER $Ca^{2+}$ release, which was abolished in *Herp*-KD astrocytes (*Figure 4K–M*), indicating the involvement of HERP in this regulation.

We then assessed whether ITPR1 and ITPR2 levels varied with time. BMAL1 phosphorylation served as a proxy for circadian phase (*Figure 4N*). ITPR2 levels showed time-dependent changes opposite to those of HERP, whereas ITPR1 levels remained constant (*Figure 4N and O*). Since *Itpr2* mRNA levels did not exhibit a rhythmic pattern (*Figure 3—figure supplement 2*), the circadian phase-dependent differences in ITPR2 levels are likely driven by HERP-mediated ITPR2 degradation. Consequently, circadian rhythmic ER $Ca^{2+}$ release is primarily driven by oscillations in ITPR2.

To verify if ER $Ca^{2+}$ release is regulated by the circadian clock, we analyzed the ER $Ca^{2+}$ response in *Bmal1$^{-/-}$* mouse astrocyte cultures. In WT astrocyte cultures, there was a clear circadian phase-dependent difference in ER $Ca^{2+}$ release. However, this difference was abolished in *Bmal1$^{-/-}$* astrocyte cultures (*Figure 5A–C*). The absence of rhythmic ER $Ca^{2+}$ release in *Bmal1$^{-/-}$* astrocytes correlated with constant levels of HERP (*Figure 5D–E*). Additionally, ER $Ca^{2+}$ release was significantly faster in *Bmal1$^{-/-}$* astrocytes compared to WT astrocytes (*Figure 5A–C*) attributed to lower levels of HERP in *Bmal1$^{-/-}$* astrocytes (*Figure 5D–E*). Collectively, these results indicate that ER $Ca^{2+}$ release is regulated by HERP, which is under circadian clock control.

## ATP-induced S368-phosphorylation of CX43 and gap junctional communication shows circadian variation

Syncytial coupling through gap junctions is a prominent feature of astrocytes, critical for their homeostatic functions, including the diffusion and equilibration of ions, metabolites, and signaling molecules (*Langer et al., 2012*; *Verkhratsky and Nedergaard, 2018*). This coupling allows the astrocytic syncytium to propagate $Ca^{2+}$ waves, affecting nearby and remote cells through release of gliotransmitters and modulating synaptic functions in a far-reaching network (*Pacholko et al., 2020*). Connexin (Cx) family proteins constitute gap junction channels (GJCs; *Giaume et al., 2021*), with Cx43 (also known as Gja1) and Cx30 (also known as Gjb6) being the main connexins in astrocytes (*Kunzelmann et al., 1999*; *Nagy et al., 1999*). In our cultured astrocytes, Cx43 was the most abundantly expressed connexin (*Figure 6—figure supplement 1*). Phosphorylation of CX43 at Ser368 by protein kinase C (PKC), activated in response to intracellular $Ca^{2+}$, is known to decrease gap junction conductance (*Enkvist and McCarthy, 1992*; *Nimlamool et al., 2015*; *Solan and Lampe, 2014*). Accordingly, we

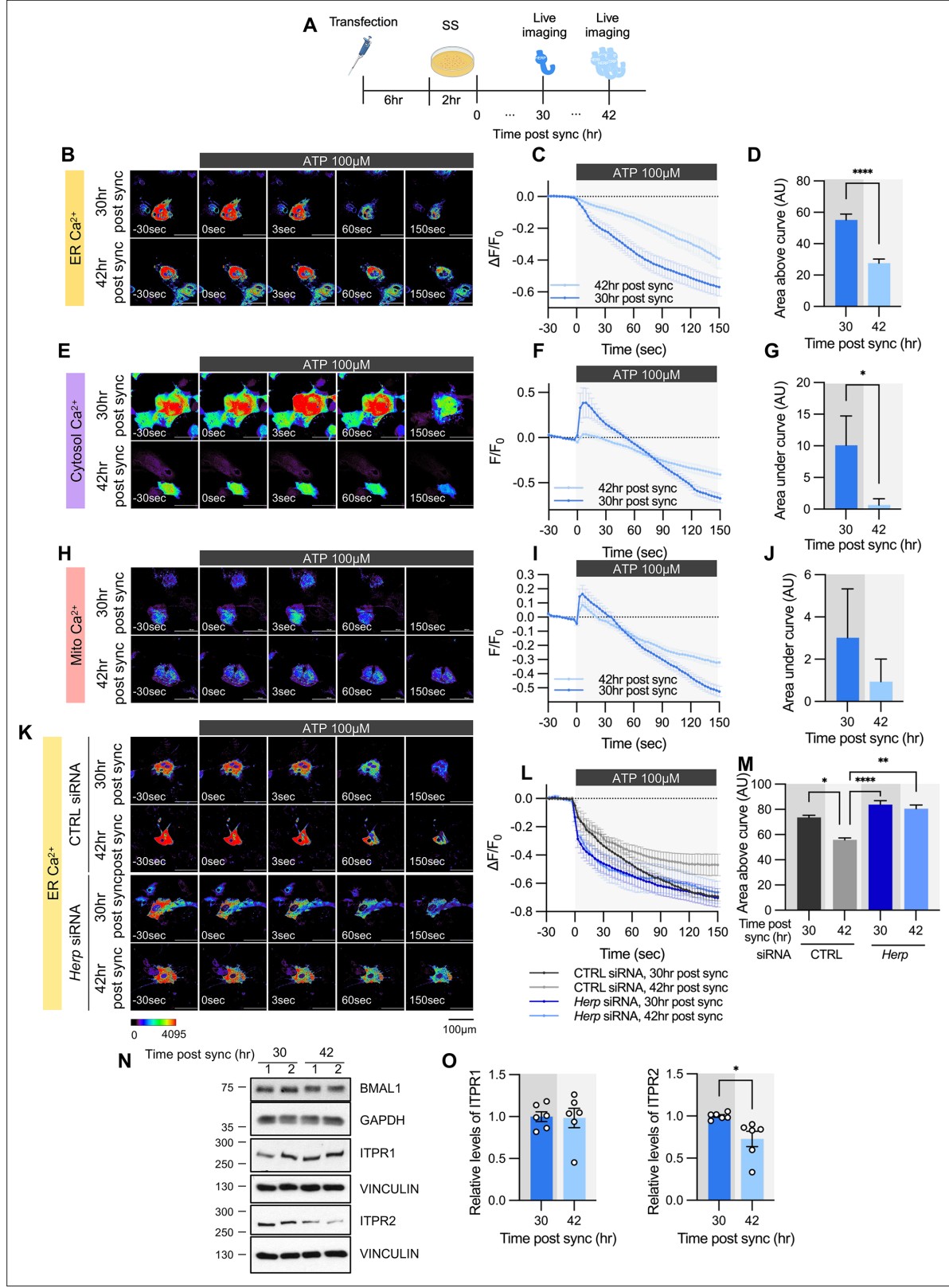

**Figure 4.** ATP-induced ER Ca²⁺ release varies according to time post sync. (**A**) Schematic diagram of the experimental scheme from transfection to live-cell Ca²⁺ imaging at different times. (**B–M**) Cultured astrocytes were transfected with G-CEPIA1er (**B–D, K–M**), R-GECO1 (**E–G**), or mito-R-GECO1 (**H–J**) compartment-specific Ca²⁺ indicators (denoted at left) and then their circadian rhythm was synchronized by SS. (**K–M**) The indicated siRNA was co-transfected with the ER Ca²⁺ indicator. After transfection, cells were allowed 48 hr for the siRNA to take effect and stabilize before synchronization by

*Figure 4 continued on next page*

*Figure 4 continued*

serum shock. At the indicated times, astrocytes were treated with 100 μM ATP and Ca²⁺ imaging was performed. (**B, E, H, K**) Representative time-lapse images of each Ca²⁺ indicator. (**C, F, I, L**) ΔF/F₀ values over time following ATP application. (**D, G, J, M**) Area above or area under the curve values, calculated from panels C, F, I, and L. (**B–D**) 30 hr post sync, n=24; 42 hr post sync, n=19. (**E–G**) 30 hr post sync, n=33; 42 hr post sync, n=38. (**H–J**) 30 hr post sync, n=50; 42 hr post sync, n=54. (**K–M**) CTRL siRNA, 30 hr post sync, n=5; CTRL siRNA, 42 hr post sync, n=4; *Herp* siRNA, 30 hr post sync, n=11; *Herp* siRNA, 42 hr post sync, n=5. Values in graphs are mean ± SEM (*p<0.05, ****p<0.00005); (**D, G, J**) *t*-test, (**M**) one-way ANOVA. (**N–O**) Cells were harvested at the indicated times and processed for western blot analysis. Vinculin and GAPDH served as loading controls for ITPR and BMAL1, respectively. (**N**) Representative western blot images from six independent experiments. (**O**) Densitometric quantification of western blot data showing relative levels of ITPR1 and ITPR2 at different times. Values in graphs are mean ± SEMs (*p<0.05, ****p<0.00005; *t*-test). Panel A was created with BioRender.com.

The online version of this article includes the following source data for figure 4:

**Source data 1.** PDF file containing original western blot for *Figure 4N*, indicating the relevant bands and treatments.

**Source data 2.** Original files for western blot analysis displayed in *Figure 4N*.

hypothesized that the circadian variations in ATP-driven ER Ca²⁺ responses might differentially impact CX43 phosphorylation according to times of day.

First, we confirmed that ATP treatment induces rapid phosphorylation of CX43 at Ser368 (pCX43 (S368)) in cultured astrocytes (*Figure 6A and B*). We then examined whether pCX43 (S368) varies

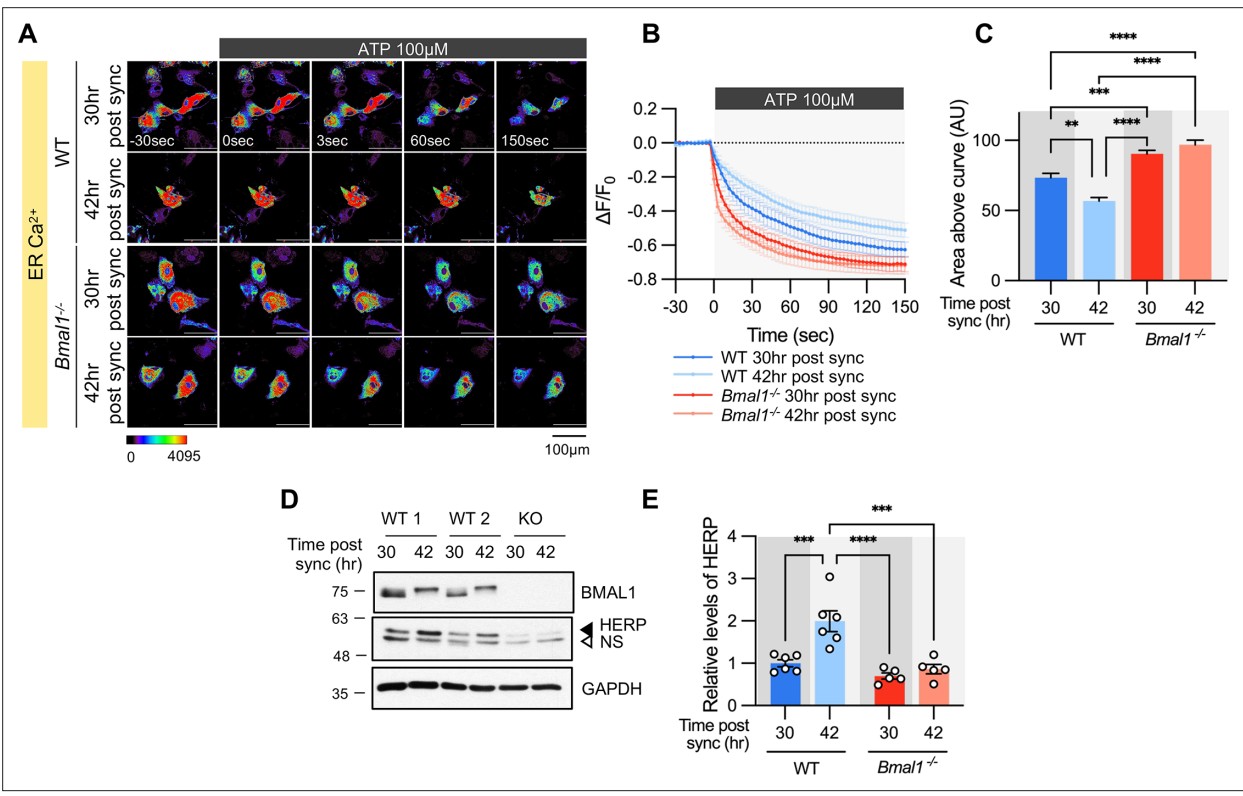

**Figure 5.** Time-dependent ER Ca²⁺ release is abolished in cultured astrocytes from *Bmal1⁻/⁻* mice. Cultured astrocytes *Bmal1⁻/⁻* mice and WT littermates were transfected with G-CEPIA1er and then their circadian rhythm was synchronized by SS. At the indicated Time, astrocytes were treated with 100 μM ATP and Ca²⁺ imaging was performed. (**A**) Representative time-lapse images of ER Ca²⁺ indicator. (**B**) ΔF/F₀ values over time following ATP application. (**C**) Area above the curve values, calculated from panel B. (**A–C**) WT 30 hr post sync, n=17; WT 42 hr post sync, n=15; KO 30 hr post sync, n=19; KO 42 hr post sync, n=21. Values are mean ± SEM (*p<0.05, **p<0.005, ***p<0.0005, ****p<0.00005; one-way ANOVA). (**D, E**) Cultured astrocytes from *Bmal1⁻/⁻* mice and WT littermates were synchronized by SS. Cells were harvested at the indicated times and processed for western blot analysis. (**D**) Representative western blot images from six independent experiments. GAPDH served as a loading control. (**E**) Values are mean ± SEM (*p<0.05, **p<0.005, ***p<0.0005, ****p<0.00005; two-way ANOVA).

The online version of this article includes the following source data for figure 5:

**Source data 1.** PDF file containing original western blot for *Figure 5D*, indicating the relevant bands and treatments.

**Source data 2.** Original files for western blot analysis displayed in *Figure 5D*.

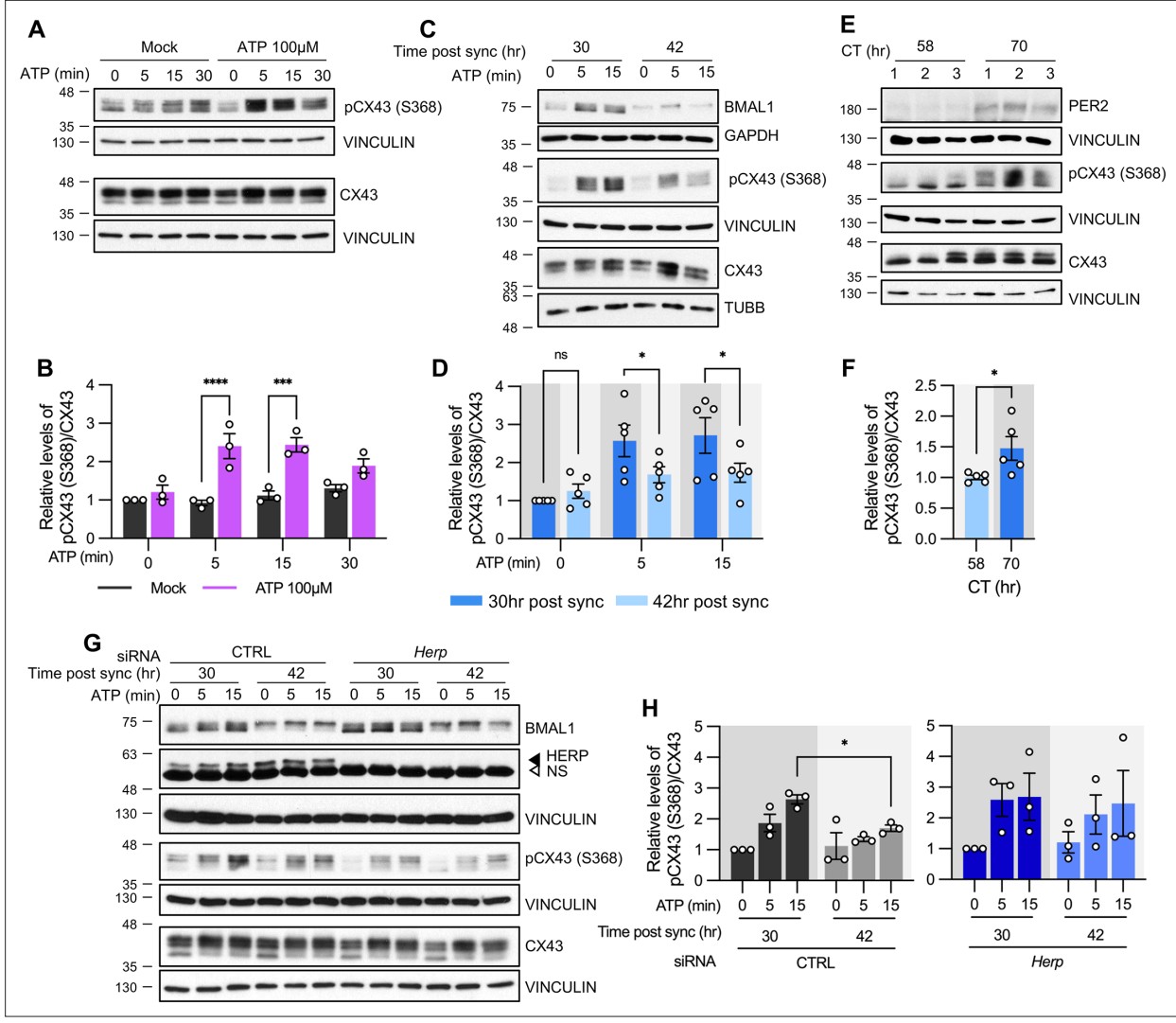

**Figure 6.** ATP-induced pCX43 (S368) levels varies according to time post sync. (**A–D, G, H**) Cultured astrocytes with (**C, D**) or without (**A, B**) synchronization, were treated with 100 μM ATP either at 30 hr post sync or 42 hr post sync and processed for western blot analysis at the indicated times. Vinculin, GAPDH, and/or β-tubulin (TUBB) served as loading controls. The intensity of pCX43 (S368) and Cx43 for each sample was normalized to that of Vinculin. (**A**) Representative western blot images from more three independent experiments. Vinculin served as a loading control. (**B**) Densitometric quantification of western blot data, showing relative pCX43 (S368)/CX43 levels. Values are normalized to those for mock-treated samples at time zero (set to 1). (**C**) Representative western blot images from three independent experiments. Vinculin, GAPDH, and β-tubulin (TUBB) served as loading controls. (**D**) Densitometric quantification of western blot data, showing relative pCX43 (S368)/CX43 levels. Values are normalized to those for mock-treated samples at ATP zero for 30 hr post sync (set to 1). (**E, F**) Changes in CX43 phosphorylation in vivo. Mice were entrained to a 12 hr light/dark cycle followed by constant dark conditions. At the indicated times, the prefrontal cortex area was dissected and processed for western blot analysis. (**E**) Representative western blot images from three independent experiments. (**F**) Densitometric quantification of western blot data showing relative levels of pCX43 (S368). (**G, H**) 48 hr post siRNA transfection, cells were synchronized by serum shock. At the indicated times post sync, 100 μM ATP was treated and processed for Western blot analysis (**G**) Representative western blot images from three independent experiments. Vinculin served as a loading control. (**H**) Densitometric quantification of Western blot data, showing relative pCX43 (S368)/CX43 levels. Values are normalized to those for 30 hr post sync at ATP zero (min) (set to 1). Values are mean ± SEM (*p<0.05, **p<0.005, ***p<0.0005, ****p<0.00005; (**B, D, H**) two-way ANOVA, (**F**) *t*-test).

The online version of this article includes the following source data and figure supplement(s) for figure 6:

**Source data 1.** PDF file containing original western blot for *Figure 6A*, indicating the relevant bands and treatments.

**Source data 2.** Original files for western blot analysis displayed in *Figure 6A*.

**Source data 3.** PDF file containing original western blot for *Figure 6C*, indicating the relevant bands and treatments.

**Source data 4.** Original files for western blot analysis displayed in *Figure 6C*.

*Figure 6 continued on next page*

*Figure 6 continued*
**Source data 5.** PDF file containing original western blot for *Figure 6E*, indicating the relevant bands and treatments.
**Source data 6.** Original files for western blot analysis displayed in *Figure 6E*.
**Source data 7.** PDF file containing Original western blot for *Figure 6G*, indicating the relevant bands and treatments.
**Source data 8.** Original files for western blot analysis displayed in *Figure 6G*.
**Figure supplement 1.** Expression profiles of Cx43 and Cx30 in cultured astrocytes from RNA-seq data.

with circadian phase. Notably, ATP-induced ER $Ca^{2+}$ release was more pronounced at 30 hr than at 42 hr post sync (*Figure 3*), and pCX43 (S368) in response to ATP was also significantly higher at 30 hr compared to 42 hr post sync (*Figure 6C and D*).

We also examined circadian variations of pCX43 (S368) in vivo. Mice were entrained to a 12:12 hr light/dark (LD) cycle for two weeks and then kept in constant darkness (DD). This ensured that variations in CX43 phosphorylation were due to the internal circadian clock and not external light stimuli. Prefrontal cortex tissues were dissected at circadian time (CT) 58 and CT70 on the third day of DD. pCX43 (S368) levels were higher at CT70 (subjective night) than at CT58 (subjective day), consistent with our in vitro results (*Figure 6E and F*).

To confirm that HERP underlies the rhythmic phosphorylation of CX43 at Ser368, we compared pCX43 (S368) in control and *Herp*-KD astrocytes at 30 hr and 42 hr post sync. In control astrocytes pCX43 (S368) was higher at 30 hr compared to 42 hr, consistent with previous results. However, in *Herp*-KD astrocytes, the circadian phase-dependent differences in pCX43 (S368) were abolished despite the normal circadian rhythm indicated by BMAL1 phosphorylation (*Figure 6G and H*). This suggests that rhythms of HERP levels are required for the rhythmic phosphorylation of CX43 (S368).

Finally, we assessed if rhythmic phosphorylation of CX43 contributes to circadian phase-dependent variation in gap junctional communication in cultured astrocytes by using fluorescence recovery after photobleaching (FRAP) method. The absence of FRAP signal in carbenoxolone (CBX)-treated astrocytes confirmed the involvement of gap junctions in intercellular communication of cultured astrocytes (*Figure 7A–C*). We observed a faster and more pronounced recovery of fluorescence in photobleached astrocytes at 42 hr compared to 30 hr post sync (*Figure 7D–F*). Importantly, this variation was absent in astrocytes cultured from $Bmal1^{-/-}$ mice, indicating that circadian clock regulates gap junctional communication (*Figure 7G–I*).

In summary, we propose a model illustrating how the circadian clock influences astrocyte function through the regulation of ER $Ca^{2+}$ response. Based on phase comparisons of clock genes between our data and CircaDB (*Figure 1D*), we defined subjective day and night, with 8 hr post sync corresponding to the beginning of the day. In astrocytes, BMAL1/CLOCK-controlled oscillation of HERP regulates day and night variations in ITPR2 levels. These oscillations contribute to changes in ER $Ca^{2+}$ responses, which in turn result in the distinct day and night differences in pCx43 (S368) and the gap junction conductance (*Figure 7J*).

## Discussion

It is widely acknowledged that the circadian clock orchestrates cellular functions in a highly cell type-specific manner by governing transcription (*Mure et al., 2018*; *Zhang et al., 2014*). Although the importance of the circadian clock in astrocytes has been well-documented (*Barca-Mayo et al., 2020*; *Barca-Mayo et al., 2017*; *Griffin et al., 2019*; *Lananna et al., 2018*; *Prolo et al., 2005*; *Tso et al., 2017*), studies specifically exploring astrocyte-specific circadian transcriptomes have been limited. Our study conducted such an analysis on primary cultured cortical astrocytes, discovering that rhythmic expression of *Herp* underlies day/night variation in ER $Ca^{2+}$ response by regulating $IP_3R$ levels. We also found that $Ca^{2+}$-activated CX43 phosphorylation varied according to the daily rhythmic ER $Ca^{2+}$ response. Given that astrocytes function in a network in which $Ca^{2+}$ propagates through GJCs, we propose that the time-dependent control of ER $Ca^{2+}$ responses could influence the role of astrocytes in modulating synaptic activity throughout the day (see below).

Circadian oscillations of *Herp* expression have been previously documented in various mouse tissues including the lung, heart, liver, kidney, aorta, skeletal muscle, adrenal gland, pancreatic islet and SCN (*Panda et al., 2002*; *Rakshit et al., 2016*; *Zhang et al., 2014*). In this study, we identified

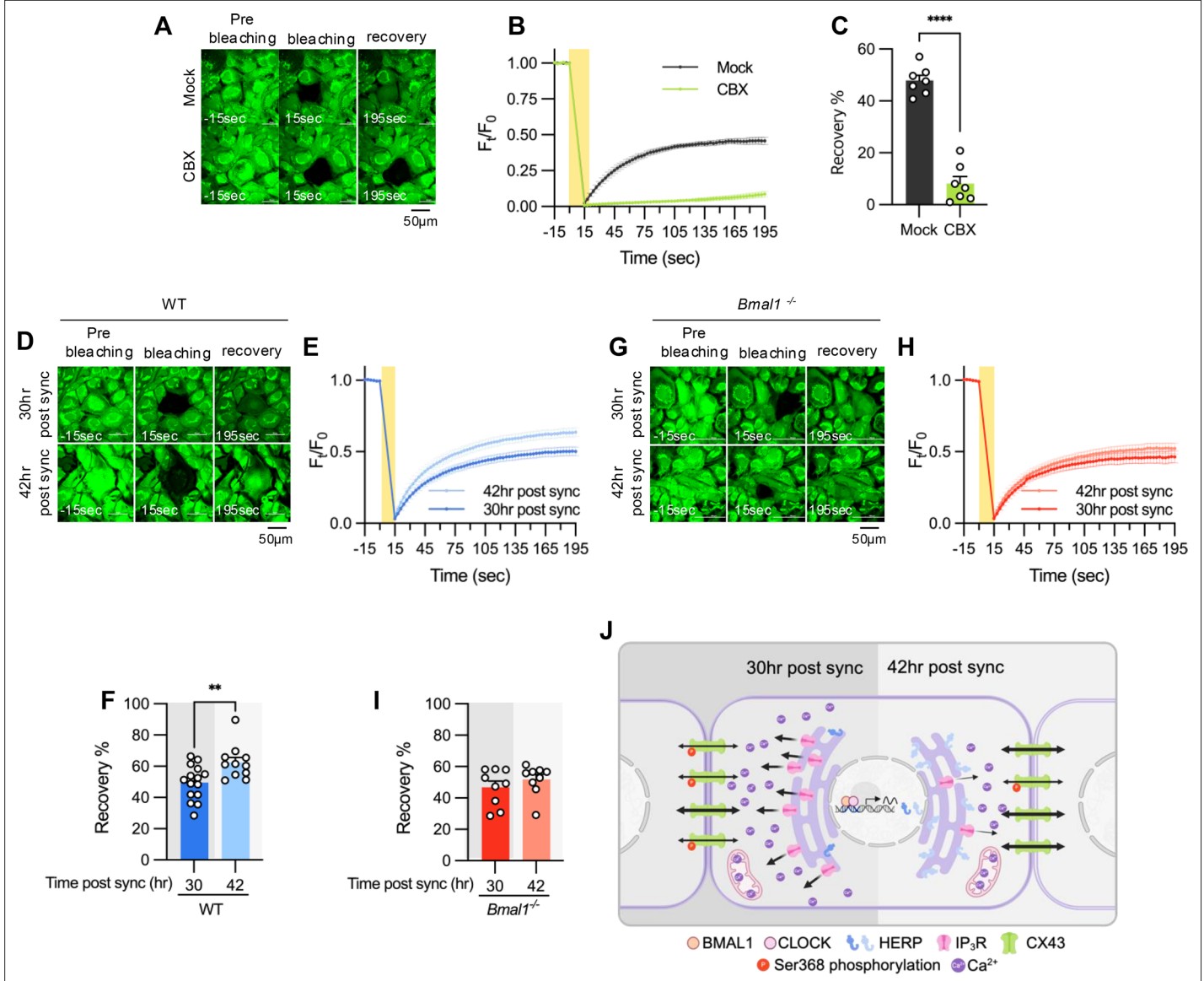

**Figure 7.** Gap junction conductance varies according to time post sync. (**A–C**) 20 µM carbenoxolone (CBX), a gap junction channel block, was applied to cultured astrocytes for 30 min before gap-FRAP analysis. (**A**) Representative time-lapse images of prebleaching, bleaching and recovery condition during gap-FRAP analysis. (**B**) $F_t/F_0$ values over time following photobleaching (yellow rectangle). (**C**) Recovery % values, calculated from panel B. Mock, n=7; CBX, n=7. Values in graphs are mean ± SEM (*p<0.05, **p<0.005, ***p<0.0005, and ****p<0.00005; Mann-Whitney $U$ test). (**D–I**) Cultured astrocytes from WT (**D–F**) and $Bmal1^{-/-}$ (**G–I**) mice were synchronized by SS. At the indicated time, gap-FRAP was performed. (**D, G**) Representative time-lapse images of prebleaching, bleaching and recovery condition during gap-FRAP analysis at 30 hr post sync and 42 hr post sync. (**E, H**) $F_t/F_0$ values over time following photobleaching (yellow rectangle). (**F, I**) Recovery % values, calculated from panel H and L, respectively. (**D–F**) 30 hr post sync, n=15; 42 hr post sync, n=11. (**G–I**) 30 hr post sync, n=9; 42 hr post sync, n=9. Values in graphs are mean ± SEM (*p<0.05, **p<0.005, ***p<0.0005, and ****p<0.00005; F: $t$-test, I: Mann-Whiteny $U$ test). (**J**) Schematic diagram illustrating the regulation of ER $Ca^{2+}$ response by the circadian clock through rhythmic oscillation of HERP. Refer to the text for the detailed explanation. Panel J was created with BioRender.com.

that *Herp* is also rhythmically expressed in cultured astrocytes. *Herp* exhibited rhythmic expression in phase with *Per2* and showed flattened expression at its nadir level in *Bmal1*⁻/⁻ astrocyte cultures, suggesting direct regulation CLOCK/BMAL1 heterodimers (*Figure 2A*). ChIP-Atlas analysis (https://chip-atlas.org; *Oki et al., 2018*; *Zou et al., 2022*) has shown CLOCK binding to the genomic region of *Herp* (*Annayev et al., 2014*; *Hong et al., 2018*). Although no canonical E-Box is present in the upstream 6 kb region of the *Herp* gene, several non-canonical E-Boxes, important for circadian expression (*Yoo et al., 2005*; *Zhang et al., 2012*), were identified (*Supplementary file 2B*). Among these,

the non-canonical E-Box sequence 'CACGTT', located near transcription start site (chr8:94386194–94386543), is highly conserved across diverse mammalian species. Therefore, we propose that these non-canonical E boxes might drive the CLOCK/BMAL1 dependent expression of *Herp*. A recent study employing a circadian single-cell RNA seq analysis of SCN reported that *Herp* displays a circadian rhythm exclusively in astrocytes and oligodendrocytes (*Wen et al., 2020*). Given that astrocytic $Ca^{2+}$ levels in the SCN show a circadian rhythm, peaking during the subjective night and are crucial for circadian timing (*Brancaccio et al., 2019*; *Patton et al., 2022*), HERP's circadian rhythmic expression in SCN astrocytes (*Wen et al., 2020*) may be integral to the mechanisms of circadian timing.

We demonstrated that HERP regulated ER $Ca^{2+}$ responses by degrading $IP_3Rs$ in astrocytes (*Figure 3J–M*). (*Paredes et al., 2016*; *Torrealba et al., 2017*). When *Herp* was downregulated, both ITPR1 and ITPR2 levels increased, suggesting HERP's role in degrading both ITPR1 and ITPR2. However, only the levels of ITPR2 exhibited circadian phase-dependent variation (*Figure 3J–M*, *Figure 4N and O*). The min/max ratio of HERP expression throughout the day was 0.38 under normal conditions but 0.072 in *Herp*-KD conditions. Based on this, we reasoned that the amplitude of HERP oscillations impacted the levels of ITPR2 under normal circumstances, while having a much lesser effect on the levels of ITPR1. Thus, the circadian control of ER $Ca^{2+}$ release in response to stimulation in cultured astrocytes is primarily attributable to ITPR2.

ER and cytosolic $Ca^{2+}$ responses following ATP treatment exhibited distinct circadian variation, but mitochondrial $Ca^{2+}$ responses did not show significant variations (*Figure 4*). We think of several possibilities to account for the absence of significant circadian variations in mitochondrial $Ca^{2+}$ relative to cytoplasmic $Ca^{2+}$ change. Upon ATP stimulation, direct $Ca^{2+}$ transfer occurs through $IP_3R$ in the cytosol. For mitochondrial uptake, $Ca^{2+}$ first transits to the intermembrane space via $IP_3Rs$ and the voltage-dependent anion channel (VDAC) complex before entering the matrix through mitochondrial calcium uniporter (MCU; *Giorgi et al., 2018*). The Mito-R-GECO1 reporter is confined within the mitochondrial matrix and thus limiting assessment of $Ca^{2+}$ levels in the intermembrane space (*Wu et al., 2013*). Substantial changes in $Ca^{2+}$ levels might occur in the intermembrane space, but the influx into the matrix via MCUs could remain constant across different time, resulting in apparent lack of significant change. Secondly, the amount of VDAC or the VDAC:$IP_3R$ complex on the mitochondrial membrane might vary inversely with HERP levels, potentially maintaining a consistent flow of $Ca^{2+}$ into the mitochondria irrespective of the time. Additionally, the presence of several $Ca^{2+}$ extrusion mechanisms in mitochondria (*Giorgi et al., 2018*) points to a more complex regulation of $Ca^{2+}$ compared with that in the cytosol. Although the precise mechanism remains to be elucidated, our data indicate that the circadian variation in $Ca^{2+}$ release from the ER, particularly in response to stimuli, predominantly impacts cytoplasmic signaling rather than mitochondrial signaling.

During the subjective night phase (active period for mice), ATP- stimulated ER $Ca^{2+}$ release was pronounced than during the subjective day phase (rest period for mice; *Figure 4*). This finding aligns with studies showing heightened stimulus-evoked $Ca^{2+}$ activity in astrocytes during wakefulness, which diminishes during sleep (*Bojarskaite et al., 2020*; *Ingiosi et al., 2020*). In flies, $Ca^{2+}$ activity in astrocytes is correlated with the duration of wakefulness, suggesting that astrocytic $Ca^{2+}$ is a key factor in the homeostatic regulation of sleep (*Blum et al., 2021*). Our findings provide a mechanistic explanation for differential $Ca^{2+}$ levels observed between wake and sleep states, with circadian regulation of ITPR2 by HERP being a key factor in this process.

ATP-stimulated Cx43 (S368) phosphorylation is higher at 30 hr (subjective night phase) than at 42 hr post sync (subjective day phase) (*Figure 6C and D*.), a finding further supported by in vivo experiments (*Figure 6E and F*). What are the implications of this day/night variation in CX43 (S368) phosphorylation? We reasoned that the circadian variation in CX43 phosphorylation could significantly impact astrocyte functionality within the syncytium. Indeed, our cultured astrocytes exhibited circadian phase-dependent variation in gap junctional communication (*Figure 7D–F*). Astrocytes influence synaptic activity through the release of gliotransmitters such as glutamate, GABA, D-serine, and ATP, triggered by increases in intracellular $Ca^{2+}$ in response to the activity of adjacent neurons and astrocytes (*Verkhratsky and Nedergaard, 2018*). Importantly, this increase in $Ca^{2+}$ spreads to adjacent astrocytes through GJCs (*Fujii et al., 2017*), influencing a large area of the neuronal network. Considering that Cx43 Ser368 phosphorylation occurs to uncouple specific pathways in the astrocytic syncytium to focus local responses (*Enkvist and McCarthy, 1992*), our findings suggest that astrocytes better equipped for localized responses when presented with a stimulus during the active phase

in mice. Conversely, during the rest period, characterized by more synchronous neuronal activity across broad brain areas (*Vyazovskiy et al., 2009*) higher GJC conductance might allow astrocytes to exert control over a larger area. In support of this idea, recent study showed that synchronized astrocytic $Ca^{2+}$ activity advances the slow wave activity (SWA) of the brain, a key feature of non-REM sleep (*Szabó et al., 2017*). Blocking GJC was found to reduce SWA, further supporting this interpretation. However, conflicting findings have also been reported. For instance, *Ingiosi et al., 2020* found that astrocytic synchrony was higher during wakefulness than sleep in the mouse frontal cortex. Whether these differing results in astrocyte synchrony during resting and active periods are attributable to differences in experimental context (e.g. brain regions, sleep-inducing condition) remains unclear. Indeed, astrocyte $Ca^{2+}$ dynamics during wakefulness/sleep vary according to brain regions (*Tsunematsu et al., 2021*). While the extent of astrocyte synchrony might differ depending on brain region and/or stimulus, our results suggest that the baseline state of astrocyte synchrony, which is affected by GJC conductance, varies with the day/night cycle.

In conclusion, our findings enhance the understanding of the circadian regulation of astrocytic functions, particularly through the rhythmic expression of HERP and its impact on ER $Ca^{2+}$ signaling and CX43 phosphorylation. This study underscores the intricate interplay between the circadian clock and astrocyte physiology, highlighting the pivotal role of HERP in modulating key astrocytic processes such as sleep, gap junction communication and synaptic modulation. The day/night variations in ER $Ca^{2+}$ response and CX43 phosphorylation, and their implications for astrocytic network dynamics and neuronal interactions, open new avenues for exploring the broader implications of astrocytic circadian rhythms in brain function and health.

# Materials and methods

## Key resources table

| Reagent type (species) or resource | Designation | Source or reference | Identifiers | Additional information |
|---|---|---|---|---|
| Gene (*Mus musculus*) | *Herpud1 (Herp)* | Mouse Genome Informatics | MGI:1927406 | |
| Gene (*M. musculus*) | *Bmal1* | Mouse Genome Informatics | MGI:1096381 | |
| Gene (*M. musculus*) | *Rev-Erbα (Nr1d1)* | Mouse Genome Informatics | MGI:2444210 | |
| Gene (*M. musculus*) | *Per2* | Mouse Genome Informatics | MGI:1195265 | |
| Gene (*M. musculus*) | *Itpr1* | Mouse Genome Informatics | MGI:96623 | |
| Gene (*M. musculus*) | *Itpr2* | Mouse Genome Informatics | MGI:99418 | |
| Gene (*M. musculus*) | *Cx43 (Gja1)* | Mouse Genome Informatics | MGI:95713 | |
| Strain, strain background (*M. musculus*, male) | C57BL/6 J | Charles River Japan | N/A | Adult PFC experiment |
| Strain, strain background (*M. musculus*, male and female) | B6.129-Bmal1^tm1Bra/J | Jackson Laboratory | Cat# 009100; RRID:IMSR_JAX:009100 | Primary Bmal1 KO astrocyte culture |
| Strain, strain background (*M. musculus*, male and female) | 1 day old C75BL/6 N | Koatch | N/A | Primary WT astrocyte culture |
| Transfected construct (*M. musculus*) | CMV-G-CEPIA1er | *Suzuki et al., 2014* | Addgene plasmid # 58215 | ER $Ca^{2+}$ indicator |
| Transfected construct (*M. musculus*) | CMV-mito-R-GECO1 | *Wu et al., 2013* | Addgene plasmid # 46021 | Mitochondria $Ca^{2+}$ indicator |
| Transfected construct (*M. musculus*) | CMV-R-GECO1 | *Zhao et al., 2011* | Addgene plasmid # 32444 | Cytosol $Ca^{2+}$ indicator |

*Continued on next page*

*Continued*

| Reagent type (species) or resource | Designation | Source or reference | Identifiers | Additional information |
|---|---|---|---|---|
| Antibody | Rabbit polyclonal anti-Bmal1 | Abcam | Cat# ab93806; RRID:AB_10675117 | 1:2000 |
| Antibody | Rabbit monoclonal anti-Herpud1 | Abcam | Cat# ab150424; RRID:AB_2857374 | 1:1000 |
| Antibody | Rabbit polyclonal anti-ITPR1 | Alomone Labs | Cat# ACC-019; RRID:AB_2039923 | 1:1000 |
| Antibody | Rabbit polyclonal anti-ITPR2 | Alomone Labs | Cat# ACC-116; Lot: ACC116AN015; RRID:AB_2340910 | 1:1000 |
| Antibody | Rabbit polyclonal anti-Connexin43 | Sigma-Aldrich | Cat# C6219; RRID:AB_476857 | 1:5000 |
| Antibody | Rabbit polyclonal anti-phospho Connexin43(Ser368) | Cell Signaling Technology | Cat# 3511 S; RRID:AB_2110169 | 1:1000 |
| Antibody | Rabbit polyclonal anti-GAPDH | Novus | Cat# NB100-56875; RRID:AB_2107610 | 1:5000 |
| Antibody | Mouse monoclonal anti-Vinculin | Sigma-Aldrich | Cat# V4505; RRID:AB_477617 | 1:5000 |
| Antibody | Rabbit polyclonal anti-Total ERK | Cell Signaling Technology | Cat# 9102 S; RRID:AB_330744 | 1:5000 |
| Sequence-based reagent | Primer used for qRT-PCR | This paper | *Supplementary file 2c*: Table of primers used in quantitative RT-PCR | |
| Sequence-based reagent | ON-TARGETplus Non-targeting siRNA negative control #1 | Horizon Discovery | Cat# D-001810–01 | 20 nM |
| Sequence-based reagent | ON-TARGETplus Mouse Herpud1 siRNA | Horizon Discovery | Cat# L-049714–01 | 20 nM |
| Sequence-based reagent | ON-TARGETplus Mouse Itpr1 siRNA | Horizon Discovery | Cat# L-040933–00 | 20 nM |
| Sequence-based reagent | ON-TARGETplus Mouse Itpr2 siRNA | Horizon Discovery | Cat# L-041018–00 | 20 nM |
| Commercial assay or kit | RNeasy Plus Kits | QIAGEN | CAT# 74034 | |
| Chemical compound, drug | Xestospongin C (XesC) | Sigma-Aldrich | Cat# X2628 | 10 μM |
| Chemical compound, drug | Adenosine 5′-triphosphate disodium salt hydrate | Sigma-Aldrich | A7699; CAS: 34369-07-8 | 100 μM |
| Chemical compound, drug | Calcein-AM | Invitrogen | C1430; CAS: 148504-34-1 | 0.5 μM |
| Chemical compound, drug | 0.05% Pluronic-127 | Invitrogen | P6866; CAS: 9003-11-6 | 0.05% |
| Chemical compound, drug | Carbenoxolone disodium salt (CBX) | Sigma-Aldrich | C4790; CAS: 7421-40-1 | 20 μM |
| Software, algorithm | ImageJ | *Schneider et al., 2012*; https://imagej.nih.gov/ij/ | RRID:SCR_003070 | |
| Software, algorithm | GraphPad Prism 7 | GraphPad Software; https://www.graphpad.com/ | RRID:SCR_002798 | |
| Software, algorithm | Nikon Element | Nikon | V5.21.00; RRID:SCR_014329 | |
| Software, algorithm | Metacycle | *Wu et al., 2016*; https://CRAN.R-project.org/package=MetaCycle | RRID:SCR_025729 | |

*Continued on next page*

Continued

| Reagent type (species) or resource | Designation | Source or reference | Identifiers | Additional information |
|---|---|---|---|---|
| Software, algorithm | BioCycle | *Agostinelli et al., 2016*; http://circadiomics.igb.uci.edu | | |
| Other | Confocal Microscope | Nikon | A1R | |
| Other | HiSeq 2000 system | Illumina | N/A | |
| Other | Metascape | *Zhou et al., 2019*; https://metascape.org | RRID:SCR_016620 | |
| Other | ChIP-Atlas | *Zou et al., 2022*; *Oki et al., 2018*; https://chip-atlas.org | RRID:SCR_015511 | |
| Other | BioRender | https://www.biorender.com/ | RRID:SCR_018361 | |

## Animals

All procedures for the care and use of laboratory animals were approved by the Institutional Animal Care and Use Committee (IACUC) of Ajou University School of medicine. *Bmal1⁻/⁺* mice (B6.129-Bmal1^tm1Bra/J; Jackson Laboratory, #009100; *Bunger et al., 2000*) were purchased from Jackson Laboratory (USA) and housed in a specific pathogen-free environment at the Animal Research Center of Ajou University Medical Center, with a standard 12 hour light/dark cycle and free access to food. Primary cultured astrocytes involving only WT mice used 1day-old C57BL/6 mice (Koatch Inc, Korea). In experiments employing primary cultured astrocytes from *Bmal1⁻/⁻* mice, wild type (WT) littermates were used as controls. For animal studies, 7–8 week-old adult C57BL/6 male mice were first entrained in a standard 12 hr light/dark cycle for 2 weeks and then maintained in a constant darkness (DD). On the third day of DD, mice were sacrificed at indicated time and its prefrontal cortex (PFC) was dissected.

## Primary astrocyte culture

Primary astrocyte culture was prepared according to a previous report (*Choi et al., 2018*) with minor modifications. In brief, 1-day-old pups were anesthetized by hypothermia and their cerebral cortices were dissected out and triturated in Modified Eagle's medium (MEM; Welgene, Korea, LM 007–11) containing 10% fetal bovine serum (FBS; Hyclone, USA, SV30207.02), penicillin/streptomycin (Gibco, USA, 15140–122), 10 μM HEPES (Gibco, USA, 15630080) and GlutaMAX (Gibco, USA, 35050–061) yielding a single-cell suspension. Cells were plated into 75 cm$^2$ T-flasks (1pup/flask) and incubated at 37 °C in a humidified 5% $CO_2$ incubator for 2 weeks. Primary astrocytes were then incubated in a serum-free MEM for 7 days, after which astrocytes were detached from the T-flask with 0.25% trypsin and plated on culture dishes (1.0X10$^6$ cells/60 mm dish) for experiments.

## Circadian transcriptome analysis

The experimental scheme for synchronizing circadian clocks of cultured astrocyte is illustrated in *Figure 1A*. In brief, astrocytes were grown in dishes (1.0X10$^6$ cells/60 mm dish) until reaching complete confluence and then incubated in serum-free MEM for 72 hr. Cells were then subjected to serum shock (SS), a well-established procedure for synchronizing the circadian clock of cultured cells (*Balsalobre et al., 1998*), by exchanging the medium for MEM containing 50% horse serum. 12 hr post SS, astrocytes were harvested at 4 hr intervals for 2 days for RNA sequencing (RNA-seq). Total RNA was extracted and purified from harvested astrocytes using a RNeasy Plus Micro Kit (QIAGEN, Germany, 74034). RNA quality and quantity assessments, RNA-Seq library construction, and next-generation sequencing analysis were performed by Macrogen Inc (Korea). Total RNA integrity and library size were analyzed using an Agilent Technologies 2100 Bioanalyzer. Paired-end raw reads, generated using an Illumina HiSeq 2000 system, were aligned to the mm10 mouse genome using

STAR aligner (v.2.5.2b) (*Dobin et al., 2013*). Gene expression was subsequently quantified by calculating transcripts per million (TPM) using Ensemble gene (release 82) annotations. The threshold for defining expressed transcripts was set using a Gaussian mixture model in the R package 'mixtools'. To detect circadian oscillating transcripts, we employed two circadian oscillation detection methods BioCycle (*Agostinelli et al., 2016*) and MetaCycle (*Wu et al., 2016*).

## Quantitative reverse transcription polymerase chain reaction (qRT-PCR)

Total RNA was extracted from cells and purified using the RNeasy Plus Micro Kit (QIAGEN, 74034). 1 µg of total RNA was reverse transcribed using an oligo-dT primer and PrimeScript RTase (TaKaRa, Japan, 2680A). Quantitative real-time PCR was performed using a Rotor Gene Q (QIAGEN) with TB Green Premix Ex Taq (Takara, RR420A). The specific primers used were provided in the *Supplementary file 2C*. Noncycling mRNA encoding HPRT was used to normalize gene expression. The data were analyzed using Rotor Gene 6000 software, and the relative mRNA levels were quantified using the $2^{-\Delta\Delta Ct}$ method in which $\Delta\Delta Ct = [(Ct_{target} - Ct_{HPRT})$ of experimental group]-[$(Ct_{target} - Ct_{HPRT})$ of control group].

## siRNA transfection and immunoblotting

The following On-Target plus SMARTpool siRNAs were purchased from Dharmacon (USA): Non-targeting CTRL siRNA (D-001810-01-50), *Herp* siRNA (L-049714-01-0005), *Itpr1* siRNA (L-040933-00-0005), and *Itpr2* siRNA (L-041018-00-0005). Upon reaching >90% confluence, the medium was replaced with serum-free MEM, and cells were incubated for 72hrs. Cells were then transfected with siRNA using Lipofectamine RNAiMAX (Thermo Fisher, USA, 13778150) per the manufacturer's instructions. Experiments were conducted 48 hrs post-transfection.

For immunoblotting, astrocytes were lysed using modified-RIPA buffer (50 mM Tris-HCl pH 7.4, 1% NP-40, 0.5% sodium deoxycholate, 150 mM NaCl). Mouse prefrontal cortex was lysed using T-Per (ThermoFisher, 78510) with protease inhibitor cocktail (Sigma-Aldrich, P8340) and phosphatase inhibitor cocktails 2 and 3 (Sigma-Aldrich, P5726 and P0044). Proteins were separated by SDS-PAGE and transferred to polyvinylidene fluoride membranes. Membranes were blocked with 5% skim milk and incubated overnight at 4°C with primary antibodies: anti-BMAL1 (Abcam (UK), ab93806), 1:2000; anti-HERP (Abcam, ab150424), 1:1000; anti-ITPR1 (Alomone Labs, Israel, ACC-019), 1:1000; anti-ITPR2 (Alomone Labs, ACC-116), 1:1000; anti-CX43 (Sigma, C6219), 1:5000; anti-pCX43 (Ser368; CST, USA, 3511), 1:1000; anti-GAPDH (Novus, USA, NB100-56875), 1:5000; anti-Vinculin (Sigma-Aldrich, V4505), 1:5000; and anti-total ERK (CST, 9102), 1:5000. Membranes were washed with TBST, incubated with secondary antibodies, and visualized using enhanced chemiluminescence. Protein levels were quantified by densitometric analysis of band intensities using ImageJ software.

## Ca²⁺ reporter plasmid transfection and imaging analysis

After cells reached >90% confluence, the medium was replaced with serum-free MEM. For $Ca^{2+}$ measurements, cells were transfected with $Ca^{2+}$ indicator plasmids pCMV-G-CEPIA1er (a gift from Masamitsu Iino, Addgene plasmid #58215; *Suzuki et al., 2014*), CMV-mito-R-GECO1, or CMV-R-GECO1 (gifts from Robert Campbell, Addgene plasmid #46021, #32444; *Wu et al., 2013*) 72 hr post-serum deprivation using Lipofectamine 2000, with or without siRNA treatment. For Xestospongin C (XesC) experiments, 10 µM XesC was applied 30 minutes before imaging. 48 hr post-transfection, the medium was replaced with $Ca^{2+}$- and $Mg^{2+}$-free HBSS (Gibco, 14175095). Fluorescence images were captured every 3 s using a Nikon A1R Confocal Microscope with a x60 1.4 NA Plan-Apochromat objective at 37°C and 5% CO2. These experiments were conducted at the Three-Dimensional Immune System Imaging Core Facility of Ajou University (Korea). Images were analyzed using NIS Elements C software. Fluorescence intensity at each time point ($F_t$) and $\Delta F$ values ($F_t$-$F_0$) were calculated. $F_0$ values were averaged from the fluorescence intensity of 10 frames prior to stimulation.

## Ca²⁺ reporter plasmid co-localization analysis

After cells reached >90% confluence, the medium was replaced with serum-free MEM. 72 hr later, cells were transfected with either the ER $Ca^{2+}$ reporter plasmid pCMV-G-CEPIA1er or the mitochondrial $Ca^{2+}$ reporter CMV-mito-R-GECO1. For ER co-localization analysis, pCMV-G-CEPIA1er was co-transfected with DsRed2-ER-5 (a gift from Michael Davidson, Addgene plasmid #55836; *Day and*

*Davidson, 2009*). For mitochondrial co-localization, 200 nM Mitotracker was applied 30 min before imaging. Live imaging was performed using a Nikon A1R Confocal Microscope with a x60 1.4 NA Plan-Apochromat objective at 37°C and 5% $CO_2$. These experiments were conducted at the Three-Dimensional Immune System Imaging Core Facility of Ajou University (Korea). Images were analyzed using NIS Elements C software.

## Gap-FRAP (fluorescence recovery after photobleaching) analysis

Astrocyte communication was evaluated using a modified gap-FRAP assay (*Santiquet et al., 2012*). 30 min before imaging, cells were incubated with 0.5 μM Calcein-AM (Invitrogen, C1430) and 0.05% Pluronic-127 (Invitrogen, P6866) at 37 °C. After incubation, cells were rinsed twice with serum-free MEM to remove excess dye. For carbenoxolone (CBX) experiments, 20 μM CBX was added for 30 min after calcein incubation. Immediately before imaging, the medium was replaced with $Ca^{2+}$- and $Mg^{2+}$-free HBSS (Gibco, 14175095). Live imaging was performed using a Nikon A1R Confocal Microscope with a 488 nm argon laser and a x60 1.4 NA Plan-Apochromat objective. Photobleaching was performed on a region of interest (ROI) cell, and fluorescence was measured at 37 °C and 5% $CO_2$. Fluorescence intensity was recorded pre-bleach (5 images), during a 15 s laser pulse (100% power, 5.3 lines/s), and post-bleach (every 3 s for 3 min). Images were analyzed using NIS Elements C software. Fluorescence intensity at each time point ($F_t$) was normalized to the pre-bleach intensity ($F_0$), calculated as the average of 5 pre-bleach frames. The percentage of fluorescence recovery was calculated using: $(F_{FR}-F_A)/(F_0-F_A) \times 100$, where $F_{FR}$ is the full recovery fluorescence intensity and $F_A$ is the intensity immediately after photobleaching.

## Quantification and statistical analysis

To conduct the statistical analyses, all samples were first tested for normality. For cases where normality testing was not possible, such as with the Area Under/Above Curve, *t*-tests were used for comparing two groups, and one-way ANOVA was used for comparing three or more groups, considering the sufficient sample size. When comparing two groups, a *t*-test was performed if the data followed a normal distribution; otherwise, the Mann-Whitney *U* test was used. For comparisons among three or more groups, one-way ANOVA was used if the data were normally distributed, and the Brown-Forsythe and Welch ANOVA tests were used if they were not. When comparing three or more groups across multiple time points, a two-way ANOVA was performed. A $p < 0.05$ was considered statistically significant for all tests. The n value for all experiments represents biological replicates, and the experiments were conducted at least three times to ensure the reliability of the results.

## Acknowledgements

We are very grateful to Eunhye Joe and all members of Eun Young Kim's laboratory for critical comments on the manuscript. This research was supported by National Research Foundation of Korea (NRF) grants funded by the Korean government (Ministry of Science and ICT; grant numbers, 2019M3C7A1031905, 2019R1A5A2026045, RS-2023-00208490, 2022R1F1A1071248).

# Additional information

### Funding

| Funder | Grant reference number | Author |
|---|---|---|
| National Research Foundation of Korea | 2019M3C7A1031905 | Eun Young Kim |
| National Research Foundation of Korea | 2019R1A5A2026045 | Eun Young Kim |
| National Research Foundation of Korea | 2022R1F1A1071248 | Jae-Hyung Lee |
| National Research Foundation of Korea | RS-2023-00208490 | Eun Young Kim |

| Funder | Grant reference number | Author |
|---|---|---|

The funders had no role in study design, data collection and interpretation, or the decision to submit the work for publication.

## Author contributions
Ji Eun Ryu, Designed and conducted experiments, analyzed the data, and wrote the manuscript; Kyu-Won Shim, Analyzed transcriptomic data and assisted with preparing the manuscript; Hyun Woong Roh, Conducted experiments, analyzed the data, and wrote the manuscript; Minsung Park, Assisted with analysis of transcriptomic data; Jae-Hyung Lee, Analyzed transcriptomic data, edited the manuscript, and acquired funding; Eun Young Kim, Designed the research, analyzed the data, wrote and edited the paper, and acquired funding

## Author ORCIDs
Ji Eun Ryu ⓘ https://orcid.org/0009-0000-1438-5018
Jae-Hyung Lee ⓘ https://orcid.org/0000-0002-5085-6988
Eun Young Kim ⓘ https://orcid.org/0000-0001-6466-8622

## Ethics
All of the animals were handled according to approved institutional animal care and use committee (IACUC) protocol of Ajou University School of medicine (permit number : 2023-0102, 2024-0103).

Reviewer #2 (Public review): https://doi.org/10.7554/eLife.96357.3.sa1
Reviewer #3 (Public review): https://doi.org/10.7554/eLife.96357.3.sa2
Author response https://doi.org/10.7554/eLife.96357.3.sa3

# Additional files

## Supplementary files
• Supplementary file 1. A total of 412 circadian rhythmic transcripts identified by MetaCycle or BioCycle, selected based on an FDR-corrected q-value <0.05.

• Supplementary file 2. Supplementary tables. (A) A table of transcripts associated with the "Calcium ion homeostasis" GO term from the GO Biological Process analysis, ordered by ascending MetaCycle p-values. (B) A comprehensive list of all non-canonical E-boxes identified in the 6 kb upstream region of Herp through non-canonical E-box analysis. (C) Primer sequences used for quantitative RT-PCR.

• MDAR checklist

## Data availability
Sequencing data have been deposited in GEO under accession codes GSE254678.

The following dataset was generated:

| Author(s) | Year | Dataset title | Dataset URL | Database and Identifier |
|---|---|---|---|---|
| Lee J, Ryu JE, Shim K, Roh HW, Park MS, Kim EY | 2024 | Circadian transcriptome analysis in mouse cultured cortical astrocytes | http://www.ncbi.nlm.nih.gov/geo/query/acc.cgi?acc=GSE254678 | NCBI Gene Expression Omnibus, GSE254678 |

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
