## [Editor Report · eLife Assessment]

This work describes a circadian regulation in the expression of HERP, a regulator of endoplasmic reticulum calcium, in primary astrocytic cultures. This work is **important** because it highlights the potential importance of circadian rhythms in astrocytes, even though making a direct comparison between these rhythms in vitro and in vivo remains challenging. The technical approaches used in this work (RNA-seq, siRNA, Ca2+ imaging) are a **solid** support for data interpretation.

---

## [Referee Report · Reviewer #2 (Public review)]

Summary:

The article by Ryu and colleagues describes the circadian control of astrocytic intracellular calcium levels in vitro.

Strengths:

The authors used a variety of technical approaches that are appropriate and considerably improved the manuscript with experiments and more solid data analysis compared to the first version

Weaknesses:

Some conceptual issues are still present. This is a mechanistic paper done completely in vitro, all references to the in vivo situation are speculative and should be absolutely avoided unless the authors are citing in vivo work.

---

## [Referee Report · Reviewer #3 (Public review)]

This study provides significant insights into how the circadian clock influences astrocytic Ca2+ homeostasis. Astrocyte biology is an active area of research and this study is timely and adds to a growing body of literature in the field. This research highlights the potential importance of circadian rhythms in astrocytes, offering a new perspective on their role in central nervous system regulation.

---

## [Author Response]

The following is the authors’ response to the original reviews.

**Public Reviews:**

**Reviewer #1 (Public Review):**
Summary:In Ryu et al., the authors use a cortical mouse astrocyte culture system to address the functional contribution of astrocytes to circadian rhythms in the brain. The authors' starting point is transcriptional output from serum-shocked culture, comparative informatics with existing tools and existing datasets. After fairly routine pathway analyses, they focus on the calcium homeostasis machinery and one gene, Herp, in particular. They argue that Herp is rhythmic at both mRNA and protein levels in astrocytes. They then use a calcium reporter targeted to the ER, mitochondria, or cytosol and show that Herp modulates calcium signaling as a function of circadian time. They argue that this occurs through the regulation of inositol receptors. They claim that the signaling pathway is clock-controlled by a limited examination of Bmal1 knockout astrocytes. Finally, they switch to calcium-mediated phosphorylation of the gap junction protein Connexin 43 but do not directly connect HERP-mediated circadian signaling to these observations. While these experiments address very important questions related to the critical role of astrocytes in regulating circadian signaling, the mechanistic arguments for HERP function, its role in circadian signaling through inositol receptors, the connection to gap junctions, and ultimately, the functional relevance of these findings is only partially substantiated by experimental evidence.Strengths:- The paper provides useful datasets of astrocyte gene expression in circadian time.- Identifies HERP as a rhythmic output of the circadian clock.- Demonstrates the circadian-specific sensitivity of ATP -> calcium signaling.- Identifies possible rhythms in both Connexin 43 phosphorylation and rhythmic movement of calcium between cells.Weaknesses:- It is not immediately clear why the authors chose to focus on Ca2+ homeostasis or Herp from their initial screens as neither were the "most rhythmic" pathways in their primary analyses.

We appreciate the reviewer’s comment. We chose to focus on Ca2+ homeostasis processes because intracellular Ca2+ signaling plays crucial role in numerous astrocyte functions and is notably associated with sleep/wake status of animals, which is our primary interest (Bojarskaite et al., 2020; Ingiosi et al., 2020; Blum et al., 2021; Szabó et al., 2017). Among the genes involved in calcium ion homeostasis, Herp exhibited the most robust rhythmicity (supplementary table 1). The rationale for our focus on Ca2+ homeostasis and Herp is explained in the results section (line 143-150). We hope this provides a clear justification for our focus.

- It would have been interesting (and potentially important) to know whether various methods of cellular synchronization would also render HERP rhythmic (e.g., temperature, forskolin, etc). If Herp is indeed relatively astrocyte-specific and rhythmic, it should be easy to assess its rhythmicity in vivo.

Thank you for the reviewer’s insightful comment. In response, we examined HERP expression in cultured astrocytes synchronized using either Dexamethasone or Forskolin treatment. We found that Herp exhibited rhythmic expression at both the the mRNA and protein levels under these conditions. These results have been added to Figure S3 and are explained in the manuscript (lines 173-175).

Additionally, we measured HERP levels in the prefrontal cortex of mice at CT58 and CT70 and found no rhythmicity, as shown in Author response image 1. Given that Herp is expressed in various brain cell types, including microglia, endothelial cells, neurons, oligodendrocytes, and the astrocytes- with the highest expression in microglia(Cahoy et al., 2008), we reason that the potential rhythmic expression of HERP in astrocytes might be masked by its continuous expression in other cell types. Nonetheless, to assess HERP rhythmicity specifically in astrocytes in vivo, we attempted immunostaining using several anti-HERP antibodies, but none were successful. Consequently, we were unable to determine whether HERP exhibits rhythmic expression in astrocytes in vivo.

**Author response image 1. sa3fig1:** HERP levels were constant at CT58 and CT70. (A, B) Mice were entrained under 12h:12h LD cycle and maintained in constant dark. Prefrontal cortices were harvested at indicated time and processed for Western blot analysis. Representative image shows three independent samples. (B) Quantification of HERP levels normalized to VINCULIN. Values in graphs are mean ± SEM (*p < 0.05, **p < 0.005, ***p < 0.0005, and ****p < 0.00005; t-test)

- The authors show that Herp suppression reduces ATP-mediated suppression of calcium whereas it initially increases Ca2+ in the cytosol and mitochondria and then suppresses it. The dynamics of the mitochondrial and cytosolic responses are not discussed in any detail and it is unclear what their direct relationship is to Herp-mediated ER signaling. What is the explanation for Herp (which is thought to be ER-specific) to calcium signaling in other organelles?

Our examination of cytosolic and mitochondrial Ca2+ responses was aimed at corroborating HERP’s effect on ER Ca2+ response. Upon ATP stimulation, Ca2+ is released from the ER via IP3R receptors (IP3Rs) and subsequently transmitted to other organelles including mitochondria (Carreras-Sureda et al., 2018; Giorgi et al., 2018). Ca2+ is directly transferred to the cytosol by IP3Rs located on the ER membrane, and to the mitochondria through a complex formed by IP3R and the voltage-dependent anion channel (VDAC) on the mitochondria (Giorgi et al., 2018). Consistent with previous reports, we observed an increase of cytosolic and mitochondrial Ca2+ levels accompanied by decrease in ER Ca2+ levels following ATP treatment (See Fig. 3B, E, H, control siRNA). The ATP-stimulated ER Ca2+ release was enhanced by Herp knockdown. We reasoned that if Ca2+ release was enhanced, then cytosolic and mitochondrial Ca2+ uptakes would also be enhanced. The results were consistent with our hypothesis (See Fig. 3B, E, H, Herp siRNA). These observations are described in the Results section (lines 202-208) and in the Discussion (lines 333-348). We hope this explanation clarifies the relationship between Herp-mediated ER Ca2+ response and Ca2+ response in other organelles. Thank you for your consideration.

- What is the functional significance of promoting ATP-mediated suppression of calcium in ER?

In astrocytes, intracellular Ca2+ plays crucial role in regulating several processes. In this study, among various downstream effects of intracellular Ca2+, we examined the gap junction channel (GJC) conductance, which affects astrocytic communication. As discussed in the manuscript (lines 357-381), circadian variation in HERP results in rhythmic Cx43 (S368) phosphorylation linked with GJC conductance. We propose that during the subjective night phase, heightened ATP induced ER Ca2+ release reduces GJC conductance, uncoupling astrocytes from the syncytium, making them better equipped for localized response. On the other hand, during the subjective day phase, increased GJC conductance may allow astrocytes to control a larger area for synchronous neuronal activity which is a key feature of sleep.

- The authors then nicely show that the effect of ATP is dependent on intrinsic circadian timing but do not explain why these effects are antiphase in cytosol or mitochondria.

Moreover, the ∆F/F for calcium in mitochondria and cytosol both rise, cross the abscissa, and then diminish - strongly suggesting a biphasic signaling event. Therefore, one wonders whether measuring the area under the curve is the most functionally relevant measurement of the change.

We appreciate the reviewer’s insightful comments. As explained in our previous response, Ca2+ released from the ER is transferred to the cytosol and mitochondria. This transfer explains why the fluorescent intensities of cytosolic and mitochondrial Ca2+ indicators show anti-phasic responses to those of the ER.

We agree that cytosolic and mitochondrial Ca2+ responses may be biphasic. The decrease below the abscissa in mitochondria and cytosol likely reflects Ca2+ extrusion from these organelles. However, our primary focus was on the initial uptake of Ca2+ following ER Ca2+ release. Thus, when calculating the area under the curve (AUC), we measured the area between the ∆F/F graph and the y=0 (X-axis) for both mitochondria and cytosol. We reason that the measuring the area under the curve (above the abscissa) fits with our objective.

While addressing your concerns, we noticed errors in the Y-axis labels of Fig. 3C, 4D, and 5C. For the ER Ca2+ dynamics, we measured the area above curve. These mistakes have now been corrected.

- Why are mitochondrial and cytosolic calcium not also demonstrated for Bmal1 KO astrocytes?

In two sets of experiments (Fig. 3 and Fig. 4), we demonstrated that the increase in cytosolic and mitochondrial Ca2+ aligns with ER Ca2+ release. Since there were no circadian time differences in ER Ca2+ release in the Bmal1 KO cultures, we concluded that it was unnecessary to measure Ca2+ levels in the mitochondria and cytosol. Additionally, our primary focus is on the ER Ca2+ response rather than the Ca2+ dynamics in subcellular organelles. We hope this clarifies our rationale and maintains the focus of our study.

- The authors claim that Herp acts by regulating the degradation of ITPRs but this hypothesis - rather central to the mechanisms proposed in this study - is not experimentally substantiated.

We appreciate the reviewer’s insightful comments regarding the role of HERP in the degradation of IP3Rs. In the original manuscript, we demonstrated that treating cells with Herp siRNA leads to an increase in the levels of ITPR1 and ITPR2, suggesting that HERP might be involved in the regulation of IP3Rs stability. This observation is consistent with previous studies, which showed that Herp siRNA treatment increases ITPR levels in HeLa and cardiac cells (Paredes et al., 2016; Torrealba et al., 2017). Torrealba et al. also showed that HERP regulates the polyubiquitination of IP3Rs. Based on our results and previous reports, we hypothesized that HERP similarly regulates ITPR degradation in cultured astrocytes.

However, as the reviewer rightly pointed out, further evidence is needed to confirm that HERP specifically regulates ITPR degradation. To address this, we conducted new experiments examining the effect of XesC, an inhibitor of IP3Rs, on ER Ca2+ release. The treatment of XesC reduced the ER Ca2+ release and abolished the enhancement of ER Ca2+ release by Herp KD. These results demonstrated that HERP influences ER Ca2+ response through IP3Rs. These new findings have been added to Fig. 3N – 3P and explained in the Results section (lines 217-221).

We believe these additional experiments and clarifications strengthen our hypothesis that HERP regulates IP3R degradation, thereby modulating ER Ca2+ responses.

- There is no clear demonstration of the functional relevance of the circadian rhythms of ATP-mediated calcium signaling.

As mentioned in the previous response, we examined Cx43 phosphorylation linked with GJC conductance in the context of ATP-mediated Ca2+ signaling. Our results demonstrated circadian variations in Cx43 Ser368 phosphorylation leading to variations of gap junction channel (GJC) conductance (Fig. 6C – F and Fig. 7D - I). We have discussed the significance of this circadian rhythm in ATP driven ER Ca2+ signaling concerning astrocytic function during sleep/wake states in the manuscript (lines 357 – 382) as follows.

“ATP-stimulated Cx43 (S368) phosphorylation is higher at 30hr (subjective night phase) than at 42hr (subjective day phase) (Fig. 6C and 6D.), a finding further supported by in vivo experiments showing higher pCx43(S368) levels in the prefrontal cortex during the subjective night than during the day (Fig. 6E and 6F). What are the implications of this day/night variation in Cx43 (S368) phosphorylation? We reasoned that the circadian variation in Cx43 phosphorylation could significantly impact astrocyte functionality within the syncytium. Indeed, our cultured astrocytes exhibited circadian phase-dependent variation in gap junctional communication (Fig.7D – 7F). Astrocytes influence synaptic activity through the release of gliotransmitters such as glutamate, GABA, D-serine, and ATP, triggered by increases in intracellular Ca2+ in response to the activity of adjacent neurons and astrocytes (Verkhratsky & Nedergaard, 2018). Importantly, this increase in Ca2+ spreads to adjacent astrocytes through GJCs (Fujii et al., 2017), influencing a large area of the neuronal network. Considering that Cx43 Ser368 phosphorylation occurs to uncouple specific pathways in the astrocytic syncytium to focus local responses (Enkvist & McCarthy, 1992), our findings suggest that astrocytes better equipped for localized responses when presented with a stimulus during the active phase in mice. Conversely, during the rest period, characterized by more synchronous neuronal activity across broad brain areas (Vyazovskiy et al., 2009) higher GJC conductance might allow astrocytes to exert control over a larger area. In support of this idea, recent study showed that synchronized astrocytic Ca2+ activity advances the slow wave activity (SWA) of the brain, a key feature of non-REM sleep (Szabó et al., 2017). Blocking GJC was found to reduce SWA, further supporting this interpretation. However, conflicting findings have also been reported. For instance, Ingiosi et al. (Ingiosi et al., 2020) found that astrocytic synchrony was higher during wakefulness than sleep in the mouse frontal cortex. Whether these differing results in astrocyte synchrony during resting and active periods are attributable to differences in experimental context (e.g., brain regions, sleep-inducing condition) remains unclear. Indeed, astrocyte Ca2+ dynamics during wakefulness/sleep vary according to brain regions (Tsunematsu et al., 2021). While the extent of astrocyte synchrony might differ depending on brain region and/or stimulus, on our results suggest that the baseline state of astrocyte synchrony, which is affected by GJC conductance, varies with the day/night cycle.”

**Reviewer #2 (Public Review):**
Summary:The article entitled "Circadian regulation of endoplasmic reticulum calcium response in mouse cultured astrocytes" submitted by Ryu and colleagues describes the circadian control of astrocytic intracellular calcium levels in vitro.Strengths:The authors used a variety of technical approaches that are appropriate

We appreciate the reviewer’s acknowledgement of the strengths of our manuscript.

Weaknesses:Statistical analysis is poor and could lead to a misinterpretation of the data

Thank you for the comment. We have carefully reviewed our statistical analyses and applied appropriate methods where necessary. Please see below for the specific revisions and improvements made.

For Fig. 2D-E, we initially used a *t*-test. However, after adding more replicates and conducting a normality test, we found that the data did not follow a normal distribution. Therefore, we switched to the Mann-Whitney *U* test. In Fig. 5D-E, we originally used a repeated measures two-way ANOVA, but we have now changed it to a standard two-way ANOVA. For Fig. 7C and I, we also observed non-normal distribution in the normality test and consequently replaced the t-test with the Mann-Whitney U test. For other analyses not specifically mentioned, normality tests confirmed normal distribution, allowing us to use t-tests or ANOVA as appropriate for statistical analysis.

Several conceptual issues have been identified.

We have addressed the reviewer’s concerns. Please see our detailed point-by-point responses below.

Overinterpretation of the data should be avoided. This is a mechanistic paper done completely in vitro, all references to the in vivo situation are speculative and should be avoided.

We appreciate the reviewer’s insightful comment. Following the reviewer’s suggestion, we have removed the interpretations of GO pathways in the context of in vivo situation.

**Reviewer #3 (Public Review):**
Astrocyte biology is an active area of research and this study is timely and adds to a growing body of literature in the field. The RNA-seq, Herp expression, and Ca2+ release data across wild-type, Bmal1 knockout, and Herp knockdown cellular models are robust and lend considerable support to the study's conclusions, highlighting their importance. Despite these strengths, the manuscript presents a gap in elucidating the dynamics of HERP and the involvement of ITPR1/2 in modulating Ca2+ release patterns and their circadian variations, which remains insufficiently supported and characterized. While the Connexin data underscore the importance of rhythmic Ca2+ release triggered by ATP, the relationship here appears correlational and the role of HERP and ITPR in Cx function remains to be characterized. Moreover, enhancing the manuscript's clarity and readability could significantly benefit the presentation and comprehension of the findings.

We appreciate the reviewer’s acknowledgement of the strengths of our manuscript. Regarding the identified gaps, we have conducted several new experiments to clearly demonstrate the HERP-ITPR-Cx phosphorylation axis. Please see our detailed point-by-point responses below.

**Recommendations for the authors:**

**Reviewer #1 (Recommendations For The Authors):**
- While HERP appears to be a clock-controlled gene and its protein levels appear to demonstrate rhythmicity as well, the data quality of the western blotting in Bmal1 knockout raises some concern about the accuracy of HERP protein quantification.

We understand the reviewer’s concern regarding the proximity of the HERP band to a nonspecific band in the Western blotting for the Bmal1 knockout. However, we took great care to ensure the accuracy of our HERP band quantification. We meticulously selected only the specific HERP band, excluding nonspecific band. Therefore, we are confident in the accuracy of our HERP protein measurements.

- If HERP is rhythmic and ITPRs are not, if their model is correct, might we expect HERP suppression to result in 'unmasking' an ITPR rhythm?

Our model suggests that both HERP and ITPRs are rhythmic, with HERP regulating the degradation of ITPR proteins and driving their rhythms. Consistent with this, we observed that day/night variations in ITPR2 levels (Fig. 4N and 4O). Therefore, we concluded that circadian variations in HERP are sufficient to drive ITPR2 rhythms. We have explained this in detail in the Result section (lines 236-241) and the Discussion section (lines 324-332).

- The authors make a rather abrupt switch to examining gap junctions and connexin 43 phosphorylation. While the data demonstrating that the phosphorylation of S368 may indeed be rhythmic - the authors do not connect these data to the rest of the manuscript by showing a connection to HERP-mediated calcium signaling, limiting the coherence of the narrative.

Thank you for the reviewer’s insightful comments. To address the reviewer's concern regarding the connection between Herp and the phosphorylation of CX43 at S368, we have conducted new experiments to test whether KD of Herp abolishes the rhythms of Cx43 phosphorylation at S368. We found that the phosphorylation of Cx43 at S368 is significantly enhanced at 30hrs post sync compared with 42hrs post sync in control siRNA-treated astrocytes consistent with our previous results (Fig. 6C & 6D). On the other hand, this circadian phase dependent difference in phosphorylation was abolished in Herp siRNA treated astrocytes. These results clearly indicate that circadian variations in Cx43 phosphorylation are driven by the HERP. These new results are now included in Fig. 6G and 6H and explained in the Results section (lines 276-281).

- Comment on data presentation: the authors repeatedly present histograms with attached lines between data points - from my understanding of the experiments, this is inappropriate unless these were repeated measures from the same cells. Otherwise, the lines connecting one data point to another between different conditions (e.g., Ctrl or Herp knockdown) are arbitrary and possibly misleading (i.e., Figure 3K, 3M, 4L, 6D).

Thank you for the reviewer’s comment. We have updated the figures by removing the lines connecting data points in the relevant figures (Fig.3K, M, Fig4.N and Fig.6D).

**Reviewer #2 (Recommendations For The Authors):**
Most of the suggestions of this reviewer are related to the conceptual interpretation and presentation of the data and to the statistical analysisIn Figure 1 the authors analyzed the rhythmic transcriptome of cortical astrocytes synchronized with a serum shock in two different ways. The authors need to discuss what is the difference between the two methods used to detect rhythmic transcripts and make sense of them.

Following the reviewer’s suggestion, we have provided a more detailed explanation about MetaCycle and BioCycle, as well as the rationale for using both packages in our analysis as follows: “Various methods have been used to identify periodicity in time-series data, such as Lomb-Scargle (Glynn et al., 2006), JTK_CYCLE (Hughes et al., 2010) and ARSER (Yang & Su, 2010), each with distinct advantages and limitations. MetaCycle, integrates these three methods, facilitating the evaluation of periodicity in time-series data without requiring the selection of an optimal algorithm (Wu et al., 2016). Additionally, BioCycle has been developed using a deep neural network trained with extensive synthetic and biological time series datasets (Agostinelli et al., 2016). Because MetaCycle and Biocycle identify periodic signal based on different algorithms, we applied both packages to identify periodicity in our time-series transcriptome data. BioCycle and MetaCycle analyses detected 321 and 311 periodic transcripts, respectively (FDR corrected, q-value < 0.05) (Fig. 1B). Among these, 220 (53.4%) were detected by both methods, but many transcripts did not overlap. MetaCycle is known for its inability to detect asymmetric waveforms (Mei et al., 2020). In our analysis, genes with increasing waveforms like Adora1 and Mybph were identified as rhythmic only by BioCycle, while Plat and Il34 were identified as rhythmic only by MetaCycle (Fig. S1C). Despite these discrepancies, the clear circadian rhythmic expression profiles of these genes led us to conclude that using the union of the two lists compensates for the limitations of each algorithm.”

Please refer to lines 105-117 in the Results section.

The reasoning for comparing CT0 with the phase of the clock 8 hs after SS needs to be explained. Circadian time (CT) conceptually refers to the clock phase in the absence of entrainment cues in vivo, the direct transformation of "time after synchronization" in vitro to CT is misleading.

Thank you for the reviewer’s insightful comments. Initially, we believed that transforming TASS to CT, despite being in vitro data, might provide a more intuitive and physiologically relevant interpretation of our results. However, we agree that this approach might be misleading. Following the reviewer’s suggestion, we have revised our terminology by changing “CT” to “Time post sync (hr)”. Nonetheless, in Fig. 1F for circular peak phase map, we set 8hrs post sync to ZT0 based on a phase comparison result in Fig. 1D for physiologically relevant interpretation. We hope these revisions clarify our approach.

Moreover, also by definition a CT cannot be defined in terms of "dark" or "light". Figure 6M needs to be changed.

Following the reviewer’s suggestion, we removed the labels CT22 and CT34. Instead. we have labeled the respective periods as “30hr post sync” and “42hr post sync”.

In Figure 1D, the authors present a gene ontology analysis that is certainly interesting, however, it should not be overinterpreted when trying to explain processes that take place only in vivo (e.g. wound repair).

Thank you for the insightful comment. Following the reviewer’s feedback, we have removed the paragraph interpreting the cell migration process in relation to wound repair and have focused instead on Ca2+ ion homeostasis.

In Figure 2A the relative expression of clock genes and Herp is again misleading by a white/grey shading indicating subjective night and subjective day when the system under study is a cell culture.

We understand the reviewer’s concern that a cell culture system is not equivalent to light/dark entrainment condition. However, we apply time-synchronizing stimuli to recapitulate in vivo entrainment. In addition, by comparing our data with CircaDB, we defined 8hrs post sync as corresponding to ZT0, thus aligning it with the beginning of the day. We have retained the shading to facilitate easier interpretation of our data in relation to in vivo situations. However, in response to the reviewer’s concern, we have revised the shading from white/grey to light grey/dark grey. We hope this adjustment addresses the reviewer’s concern, but if the reviewer still believes it is inappropriate, please let us know, we will gladly update it.

In the Figure 2A legend, it is indicated that rhythmicity is assessed using MetaCycle with mean values obtained from n=2. The authors need to make clear whether this n=2 mean: 2 biological replicates or 2 technical replicates. This difference is relevant because it would make the analysis statistically valid or invalid, respectively.

Thank you for your feedback. n=2 refers to 2 biological replicates. Therefore, the analysis is statistically valid.

In Figures 2C and D the authors applied a T-test, a parametric statistical test for one-to-one comparison that requires normality distribution of the data to be tested first. To test normality, the authors need at least 4 biological replicates. The suggestion of this reviewer is that these experiments have to be repeated and proper statistics applied.

Thank you for your feedback. In response to the reviewer's suggestion, we conducted additional experiments to increase the number of biological replicates to 4. After verifying the normality of the data, we applied a t-test for Figure 2C and a Mann-Whitney test for Figure 2D and 2E. These tests confirmed significant statistical difference between groups.

Further evidence of Bmal1-dependent control of HERP circadian expression authors could check the presence of E-Box elements in the Herp promoter.

Thank you for the reviewer’s insightful comment. In the original version of our manuscript's Discussion section, we mentioned the absence of a canonical E-Box in the upstream of Herp gene. However, following the reviewer’s suggestion and considering the potential role of non-canonical E-Boxes, we conducted an additional analysis. This analysis identified several non-canonical E-Boxes within the 6 kb upstream region of the Herp gene (Table S2). Notably, we found one non-canonical E-Box, “CACGTT,” known to regulate circadian expression (Yoo et al., 2005) is close to the transcription start site (chr8:94386194-94386543). Moreover, this element is evolutionarily conserved across various mammals, including humans, rats, mice, dogs, and opossums (See Author response image 2). Therefore, we reasoned that these non-canonical E boxes might drive the CLOCK/BMAL1 dependent expression of Herp. We have updated the Discussion to reflect these findings in lines 315-319.

**Author response image 2. sa3fig2:** 

The calcium experiments shown in Figures 3A-I, could be more convincing if the authors showed that the different Ca2+ sensors are compartment-specific by showing co-localization with a subcellular marker. In the pictures shown it is not even possible to recognize the cell dimensions.

Following the reviewer’s suggestion, we performed co-staining experiments with organelle specific Ca2+ indicators and organelle markers. First, astrocytes were co-transfected with G-CEPIA1er, an ER specific Ca2+ indicator and ER targeted DsRed2 (with Calreticulin signal sequence). Live imaging analysis showed that the fluorescent intensities of G-CEPIA1er and DsRed2-ER-5 significantly overlapped in co-transfected cells. Secondly, astrocytes were transfected with Mito-R-GECO1 and Mitotracker, a cell permeable mitochondria dye, was applied. The fluorescent intensities of Mito-R-GECO1 and Mitotracker also significantly overlapped. These new data are included in Figure S4 and explained in the Result section (lines 194-195).

Data analysis in Figure 3 K and M is misleading. According to the explanations of the results, each of the experiments to assess ITRP1 or 2 is run independently. Then it is not clear why the relative levels obtained with control or Herp siRNA are plotted as pairs. Same comment as above for Figure 4L and Figure 6D.

Thank you for the reviewer’s insightful comments. Reviewer1 raised similar issues. Following the reviewers’ suggestions, we have removed the lines connecting the data points in Fig. 3K, 3M, 4L, and 6D.

In Figure 5E the authors need to explain why they consider that repeated measures 2-way ANOVA is the right statistical test to apply. According to the explained experimental design, cells transfected, synchronized, and then harvested independently at the indicated time after synchronization.

Thank you for the reviewer’s insightful comment. Upon reviewing the statistical methods as suggested, we have revised our approach. Instead of using repeated measures 2-way ANOVA, we have now applied a standard 2-way ANOVA, which is more appropriate given the experimental procedures were independent, as the reviewer pointed out.

The English language needs to be revised throughout the text.

We have thoroughly revised the English language throughout the text.

**Reviewer #3 (Recommendations For The Authors):**
(1) Figure 3. Clarify the physiological importance of 100 µM ATP. Would the Herp rhythm warrant Ca2+ release rhythms under basal conditions? In 3J-K, the relatively weak effect of Herp knockdown on ITPR1/2 levels, albeit statistically significant, may not be physiologically significant. This calls into question the claimed Herp-ITPR axis that underlies the Ca2+ release phenotype. Further, the correlation certainly exists but further characterization of Herp KD cells would be required to address the mechanism.

As previously reported, a broad range of ATP concentrations can induce Ca2+ activity in the astrocytes (Neary et al., 1988). Originally, we conducted an ATP dose-response analysis to observe ER Ca2+ release in our primary astrocyte culture. Our results show that ER Ca2+ release begins at 50 µM ATP and plateaus at 500 µM. Please refer to Author response image 3. We selected 100µM ATP for our experiments because it induces a medium level of ER Ca2+ response. Importantly, although measuring ATP concentrations at the synapse in vivo is challenging(Tan et al., 2017), estimates suggest synaptic ATP concentrations range from 5-500 µM (Pankratov et al., 2006). Thus, 100µM ATP is a physiologically relevant concentration that can affect nearby cells, including astrocytes, in the nervous system.

**Author response image 3. sa3fig3:** Cultured astrocytes were transfected with G-CEPIA1er ER and at 48hrs post transfection, cultured astrocytes were treated with various concentrations of ATP and Ca2+ imaging analysis was performed. (A) ΔF/F0 values over time following ATP application. (B) Area above curve values. Values in graphs are mean ± SEM (*p < 0.05, **p < 0.005, ***p < 0.0005, and ****p < 0.00005; one-way ANOVA).

Regarding the comment on Ca2+ release rhythms under basal conditions, we interpret this as referring Ca2+ release in the absence of a stimulus. We typically observe Ca2+ release only upon stimulation, such as ATP treatment. However, we acknowledge that the modest effects of HERP knockdown on ITPR1/2 levels could question the HERP-ITPR axis’s role in ER Ca2+ release.

To address this, we analyzed whether Herp KD induced increases in ER Ca2+ release were mediated through ITPRs by treating cells with Xestospongin C (XesC), an IP3R inhibitor. XesC treatment reduced ATP-induced ER Ca2+ release and eliminated the differences in ER Ca2+ release between control and Herp KD astrocytes (Fig. 3N – 3P). These results clearly indicate that HERP-ITPR axis plays critical role in controlling ER Ca2+ release. These new experiments have been included in Fig. 3 and explained in the result section (lines 217-221).

Furthermore, following the reviewer’s suggestion, we examined whether HERP rhythms underlie the rhythms of ER Ca2+ response by analyzing ER Ca2+ response in Herp KD astrocyte in two different times following synchronization. In control astrocytes, ATP-induced ER Ca2+ responses vary depending on time, whereas these time-dependent variations were abolished in Herp KD astrocytes. These new experiments have been included in Fig. 4K – 4M and explained in the Results section (lines 232-235).

Collectively, these results indicate that HERP rhythms lead to time-dependent differences in ER Ca2+ response through ITPRs.

(2) Figure 4K-L. As data suggested the involvement of ITPR1 and ITPR2 (circadian effect), a reasonable next step is to determine their involvement, but the study did not pursue the hypothesis.

Thank you for your insightful comment. Our results indeed suggest that rhythms in ITPR2 levels may drive the time-dependent variations in ATP-induced ER Ca2+ release following synchronization. The newly conducted experiments demonstrated that treatment with the ITPR inhibitor XesC suppressed ATP-induced ER Ca2+ release at both control and Herp siRNA treatment conditions (Fig. 3). Based on these findings, we now further confirm that rhythms of ITPR levels, specifically ITPR2 underlie the circadian variations in ER Ca2+ release. While examining the effect of ITPR2 siRNA would directly prove the involvement of ITPR2, we have decided to pursue this experiment in the future studies.

(3) Figure 5A-C. Data from WT cells should be included side by side with Bmal1-/- cells for comparison which is expected to be consistent with the HERP levels as in 5D-E. Again, the role of ITPR2 is suggested but not demonstrated.

Following the reviewer's suggestion, we conducted additional experiments including both WT and Bmal1-/- cultured astrocytes side-by-side. The results were consistent with our previous findings: WT astrocytes showed rhythms of ER Ca2+ release while Bmal1-/- astrocytes did not. We have updated the Figure 5A to 5C and the corresponding Results section in lines 242-245 accordingly.

Regarding second comment, as mentioned in our previous response, we plan to examine the role of ITPR2 in further studies.

(4) Figure 6. The Connexin data seems an addon and is correlative with the Ca2+ release. The role of Herp and Itpr in Connexin function is not addressed. Figure 6E-F was not called out in the results section. Suggest providing additional data to support the role of the HERP-ITPR axis in regulating Ca2+ release and Connexin activity.

We agree that additional data are needed to support the role of HERP in regulating CX43 phosphorylation. Therefore, we have conducted further experiments to determine whether rhythms of Cx43 phosphorylation are regulated by HERP. In the control astrocytes, ATP treatment induced time-dependent variations in Cx43 phosphorylation. However, these rhythms were abolished in Herp KD astrocytes. These results indicate that rhythms in HERP levels contribute to the time-dependent variations in Cx43 phosphorylation. These new experiments have included in Fig. 6G and 6H and explained in the results section (lines 276-281).

Regarding second comment, we have corrected our oversight by properly referencing figures 6E-F in the results section. Please refer to lines 357-359 for clarification.

(5) Discussion. This section should focus on noteworthy points to discuss, not repeating the results.

Based on the reviewer's valuable suggestions, we have revised the Discussion section to minimize repetition of the results. Thank you for your guidance.

(6) The manuscript exhibits numerous grammatical and textual inaccuracies that necessitate careful revision by the authors. My observations here are confined to the title and the abstract alone. I recommend altering the title from "mouse cultured astrocytes" to "cultured mouse astrocytes" for clarity and grammatical correctness. The abstract, meanwhile, needs enhancements both in terms of its content and language. It should incorporate the results of the partitioning among the ER, cytoplasm, and mitochondria, and provide clear definitions for some of the critical terms used. It's worth noting that the abstract's second sentence contains a grammatical error.

Thank you for the reviewer’s valuable feedback. We have carefully revised the title, abstract, and main text to address the grammatical and textual issues. The title has been changed to “cultured mouse astrocytes”. Additionally, the abstract now includes results related to cytoplasmic Ca2+ dynamics and has been revised in several places. We appreciate your insights and have worked to enhance the content and language accordingly.

Reference

Agostinelli, F., Ceglia, N., Shahbaba, B., Sassone-Corsi, P., & Baldi, P. (2016). What time is it? Deep learning approaches for circadian rhythms. *Bioinformatics*, *32*(12), i8-i17. https://doi.org/10.1093/bioinformatics/btw243

Cahoy, J. D., Emery, B., Kaushal, A., Foo, L. C., Zamanian, J. L., Christopherson, K. S., Xing, Y., Lubischer, J. L., Krieg, P. A., Krupenko, S. A., Thompson, W. J., & Barres, B. A. (2008). A transcriptome database for astrocytes, neurons, and oligodendrocytes: a new resource for understanding brain development and function. *J Neurosci*, *28*(1), 264-278. https://doi.org/10.1523/JNEUROSCI.4178-07.2008

Carreras-Sureda, A., Pihán, P., & Hetz, C. (2018). Calcium signaling at the endoplasmic reticulum: fine-tuning stress responses. *Cell Calcium*, *70*, 24-31. https://doi.org/10.1016/j.ceca.2017.08.004

Enkvist, M. O., & McCarthy, K. D. (1992). Activation of protein kinase C blocks astroglial gap junction communication and inhibits the spread of calcium waves. *J Neurochem*, *59*(2), 519-526. https://doi.org/10.1111/j.1471-4159.1992.tb09401.x

Fujii, Y., Maekawa, S., & Morita, M. (2017). Astrocyte calcium waves propagate proximally by gap junction and distally by extracellular diffusion of ATP released from volume-regulated anion channels. *Scientific Reports*, *7*(1), 13115. https://doi.org/10.1038/s41598-017-13243-0

Giorgi, C., Marchi, S., & Pinton, P. (2018). The machineries, regulation and cellular functions of mitochondrial calcium. *Nature Reviews Molecular Cell Biology*, *19*(11), 713-730. https://doi.org/10.1038/s41580-018-0052-8

Glynn, E. F., Chen, J., & Mushegian, A. R. (2006). Detecting periodic patterns in unevenly spaced gene expression time series using Lomb-Scargle periodograms. *Bioinformatics*, *22*(3), 310-316. https://doi.org/10.1093/bioinformatics/bti789

Hughes, M. E., Hogenesch, J. B., & Kornacker, K. (2010). JTK_CYCLE: an efficient nonparametric algorithm for detecting rhythmic components in genome-scale data sets. *J Biol Rhythms*, *25*(5), 372-380. https://doi.org/10.1177/0748730410379711

Ingiosi, A. M., Hayworth, C. R., Harvey, D. O., Singletary, K. G., Rempe, M. J., Wisor, J. P., & Frank, M. G. (2020). A Role for Astroglial Calcium in Mammalian Sleep and Sleep Regulation. *Curr Biol*, *30*(22), 4373-4383.e4377. https://doi.org/10.1016/j.cub.2020.08.052

Mei, W., Jiang, Z., Chen, Y., Chen, L., Sancar, A., & Jiang, Y. (2020). Genome-wide circadian rhythm detection methods: systematic evaluations and practical guidelines. *Briefings in Bioinformatics*, *22*(3). https://doi.org/10.1093/bib/bbaa135

Neary, J. T., van Breemen, C., Forster, E., Norenberg, L. O., & Norenberg, M. D. (1988). ATP stimulates calcium influx in primary astrocyte cultures. *Biochem Biophys Res Commun*, *157*(3), 1410-1416. https://doi.org/10.1016/s0006-291x(88)81032-5

Pankratov, Y., Lalo, U., Verkhratsky, A., & North, R. A. (2006). Vesicular release of ATP at central synapses. *Pflugers Arch*, *452*(5), 589-597. https://doi.org/10.1007/s00424-006-0061-x

Paredes, F., Parra, V., Torrealba, N., Navarro-Marquez, M., Gatica, D., Bravo-Sagua, R., Troncoso, R., Pennanen, C., Quiroga, C., Chiong, M., Caesar, C., Taylor, W. R., Molgó, J., San Martin, A., Jaimovich, E., & Lavandero, S. (2016). HERPUD1 protects against oxidative stress-induced apoptosis through downregulation of the inositol 1,4,5-trisphosphate receptor. *Free Radic Biol Med*, *90*, 206-218. https://doi.org/10.1016/j.freeradbiomed.2015.11.024

Szabó, Z., Héja, L., Szalay, G., Kékesi, O., Füredi, A., Szebényi, K., Dobolyi, Á., Orbán, T. I., Kolacsek, O., Tompa, T., Miskolczy, Z., Biczók, L., Rózsa, B., Sarkadi, B., & Kardos, J. (2017). Extensive astrocyte synchronization advances neuronal coupling in slow wave activity in vivo. *Scientific Reports*, *7*(1), 6018. https://doi.org/10.1038/s41598-017-06073-7

Tan, Z., Liu, Y., Xi, W., Lou, H. F., Zhu, L., Guo, Z., Mei, L., & Duan, S. (2017). Glia-derived ATP inversely regulates excitability of pyramidal and CCK-positive neurons. *Nat Commun*, *8*, 13772. https://doi.org/10.1038/ncomms13772

Torrealba, N., Navarro-Marquez, M., Garrido, V., Pedrozo, Z., Romero, D., Eura, Y., Villalobos, E., Roa, J. C., Chiong, M., Kokame, K., & Lavandero, S. (2017). Herpud1 negatively regulates pathological cardiac hypertrophy by inducing IP3 receptor degradation. *Sci Rep*, *7*(1), 13402. https://doi.org/10.1038/s41598-017-13797-z

Tsunematsu, T., Sakata, S., Sanagi, T., Tanaka, K. F., & Matsui, K. (2021). Region-specific and state-dependent astrocyte Ca^2+^ dynamics during the sleep-wake cycle in mice. *The Journal of Neuroscience*, JN-RM-2912-2920. https://doi.org/10.1523/jneurosci.2912-20.2021

Verkhratsky, A., & Nedergaard, M. (2018). Physiology of Astroglia. *Physiol Rev*, *98*(1), 239-389. https://doi.org/10.1152/physrev.00042.2016

Vyazovskiy, V. V., Olcese, U., Lazimy, Y. M., Faraguna, U., Esser, S. K., Williams, J. C., Cirelli, C., & Tononi, G. (2009). Cortical firing and sleep homeostasis. *Neuron*, *63*(6), 865-878. https://doi.org/10.1016/j.neuron.2009.08.024

Wu, G., Anafi, R. C., Hughes, M. E., Kornacker, K., & Hogenesch, J. B. (2016). MetaCycle: an integrated R package to evaluate periodicity in large scale data. *Bioinformatics*, *32*(21), 3351-3353. https://doi.org/10.1093/bioinformatics/btw405

Yang, R., & Su, Z. (2010). Analyzing circadian expression data by harmonic regression based on autoregressive spectral estimation. *Bioinformatics*, *26*(12), i168-174. https://doi.org/10.1093/bioinformatics/btq189

Yoo, S. H., Ko, C. H., Lowrey, P. L., Buhr, E. D., Song, E. J., Chang, S., Yoo, O. J., Yamazaki, S., Lee, C., & Takahashi, J. S. (2005). A noncanonical E-box enhancer drives mouse Period2 circadian oscillations in vivo. *Proc Natl Acad Sci U S A*, *102*(7), 2608-2613. https://doi.org/10.1073/pnas.0409763102